# Holocene wildfire regimes in western Siberia: interaction between peatland moisture conditions and the composition of plant functional types

Angelica Feurdean*[1,2,3], Andrei-Cosmin Diaconu[4], Mirjam Pfeiffer[2], Mariusz Gałka[5], Simon M. Hutchinson[6], Geanina Butiseaca[2], Natalia Gorina[7], Spassimir Tonkov[8], Aidin Niamir,[2] Ioan Tantau[4], Hui Zhang[9] Sergey Kirpotin[10,11]

[1]Department of Physical Geography, Goethe University, Altenhöferallee 1, 60438 Frankfurt am Main, Germany
[2]Senckenberg Biodiversity and Climate Research Centre (SBiK-F), Senckenberganlage, 25, 60325, Frankfurt am Main, Germany
[3] STAR-UBB Institute, Babeș-Bolyai University, Kogălniceanu 1, 400084, Cluj-Napoca, Romania
[4]Department of Geology, Babeș-Bolyai University, Kogălniceanu 1, 400084, Cluj-Napoca, Romania
[5]Department of Biogeography, Paleoecology and Nature Conservation, Faculty of Biology and Environmental Protection,
University of Lodz, Banacha 1/3, Lodz, Poland
[6]School of Science, Engineering and Environment, University of Salford, Greater Manchester M5 4WT, Salford, UK
[7]Department of Ecology, Natural Use and Environmental Engineering, National Tomsk State University, Lenina Pr., 36, 634050, Tomsk, Russia
[8]Laboratory of Palynology, Faculty of Biology, Sofia University St. Kliment Ohridski, Dragan Tsankov 8, 1164, Sofia, Bulgaria
garia
[9]Key Laboratory of Cenozoic Geology and Environment, Institute of Geology and Geophysics, Chinese Academy of Sciences, No. 19, Beitucheng Western Road, Chaoyang District, 100029, Beijing, China
[10]Tuvan State University, Lenina 36, 667000, Kyzyl, Russia
[11]Bio-Clim-Land Center of Excellence, National Research, Tomsk State University, Lenina Pr., 36, Tomsk, 634050, Russia

*Correspondence to*: Feurdean@em.uni-frankfurt.de; angelica.feurdean@gmail.com


**Abstract.** Wildfire is the most common disturbance type in boreal forests and can trigger significant changes in forest composition. Waterlogging in peatlands determines the degree of tree cover and the depth of the burnt horizon associated with wildfires. However, interactions between peatland moisture, vegetation composition and flammability, and fire regime in forest

and forested peatland in Eurasia remain largely unexplored, despite their huge extent in boreal regions. To address this knowledge gap, we reconstructed the Holocene fire regime, vegetation composition and peatland hydrology at two sites located in predominantly light taiga (*Pinus sylvestris-Betula*) with interspersed dark taiga communities (*Pinus sibirica, Picea obovata, Abies sibirica)* in Western Siberia in the Tomsk Oblast, Russia. We found marked shifts in past water levels over the Holocene. The probability of fire occurrence and the intensification of fire frequency and severity increased at times of low water table

(drier conditions), enhanced fuel dryness, and an intermediate dark-to-light-taiga ratio. High water level, thus wet peat surface conditions, prevented fires from spreading on peatland and surrounding forests. Deciduous trees (i.e., *Betula*) and *Sphagnum*, were more abundant under wetter peatland conditions, and conifers and denser forests were more prevalent under drier peatland conditions. On a Holocene scale, severe fires were recorded between 7.5 and 4.5 ka BP with an increased proportion of dark taiga and fire avoiders (*Pinus sibirica* at Rybnaya and *Abies sibirica* at Ulukh-Chayakh) in a predominantly light taiga and

fire-resister community characterized by *Pinus sylvestris* and lower local water level. Severe fires also occurred over the last 1.5 ka BP and were associated with a declining abundance of dark taiga, fire avoiders, an expansion of fire invaders (*Betula*), and fluctuating water tables. These findings suggest that frequent, high-severity fires can lead to compositional and structural changes in forests when trees fail to reach reproductive maturity between fire events or where extensive forest gaps limit seed dispersal. This study also shows prolonged periods of synchronous fire activity across the sites, particularly during the early

to mid-Holocene, suggesting a regional imprint of centennial to millennial-scale Holocene climate variability on wildfire activity. Humans may have affected vegetation and fire from the Neolithic, however, increasing human presence in the region and particularly at the Ulukh-Chayakh Mire over the last four centuries drastically enhanced ignitions compared to natural background levels. Frequent warm and dry spells predicted by climate change scenarios for Siberia in the future will enhance peatland drying and may convey a competitive advantage to conifer taxa. However, dry conditions will probably exacerbate

the frequency and severity of wildfire, disrupt conifers' successional pathway and accelerate shifts towards deciduous broadleaf tree cover. Furthermore, climate-disturbance-fire feedbacks will accelerate changes in the carbon balance of boreal peatlands and affect their overall future resilience to climate change.

## 1 Introduction

Wildfire is the most common type of disturbance in boreal forests and forested peatlands (Kharuk et al., 2021 and refs therein).

It can change tree community composition and accelerate climate warming via carbon release into the atmosphere and alteration of the radiative balance due to changes in land surface albedo (Rogers et al., 2015; Kharuk et al., 2021). However, the

impacts of wildfire on vegetation and climate strongly depend on the fire regime. High-intensity crown fires kill most trees and alter species composition for an extended period of time. In the short term, such fires release black carbon aerosols (Rogers et al., 2015). When these aerosols persist in the atmosphere, this leads to a medium-term increase in albedo and ultimately to regional cooling. Contrastingly, surface fires typically do not kill mature trees or trigger stand-scale forest replacement and have little effect on albedo. Fire types in the Siberian boreal forests and forested peatlands are often litter-fuelled surface fires that infrequently transition to the crown, depending on forest composition (Sannikov and Goldammer, 1996; Grooth et al., 2013; Kharuk et al., 2021). A high diversity of fire-related plant functional types (PFTs), including resisters (*Pinus sylvestris, Larix sibirica, L. gmelinii*), avoiders (*Abies sibirica, Picea obovata, Pinus sibirica*), invaders (*Betula pubescens, B. pendula*) and endurers (*Populus tremula, B. pubescens*) dominate these forest communities (Rowe, 1983; Agee, 1998; Wirth, 2005). Fire resisters, invaders, and endurers are commonly associated with high-frequency but low-intensity surface fires, whereas avoiders are associated with low-frequency, but high-intensity crown fires (Goldammer and Furyaev, 1996; Wirth, 2005). As these two types of fire can have a different net effect on tree community composition and climate, it is essential to understand the patterns and drivers behind each fire regime. This understanding will also improve the prediction of the impact of changes in the extent, frequency, and severity of fire in the future.

A fire regime emerges from the combination of ignition sources, climatic conditions, fuel properties and human activities. The interactions between these factors are spatiotemporally complex (Moritz et al., 2014; Andela et al., 2017). The spatial complexity of wildfire patterns is particularly accentuated in peatlands, where the variability in the local peat moisture content, vegetation composition and structure results in pronounced small-scale heterogeneity (Camill et al., 2009; Magnan et al., 2012; Kuosmanen et al., 2014; Remy et al., 2018; Barhoumi et al., 2019; Stivrins et al., 2019; Feurdean et al., 2020a). The waterlogged conditions that prevail in peatlands can limit the depth to which the fuel is burnt. However, this attenuation effect decreases with water table decline and increasing tree cover (Whitman et al., 2018). Unlike drained peatlands with shrub and tree-dominated communities, well-hydrated peatlands composed of a dense *Sphagnum* cover remain wet despite droughts and exhibit only slight to medium fire damage (Whitman et al., 2018; Gewin, 2020). This pattern in fire dynamics is particularly concerning because climate warming accelerates woody encroachment in peatlands and increases fuel availability for peatland forest fires (Kharuk et al., 2021). Intensification of fire severity may also lead to a shift from conifer towards deciduous tree dominance (Kelly et al., 2013; Mekonnen et al., 2019). Understanding how hydrological conditions in peatlands are influenced by climate change and how this may interact with forest composition and fuel flammability is a key research priority that remains largely unexplored (Page et al., 2009; Walker et al., 2009; Kharuk et al., 2021).

Reducing uncertainty in predicting the future trajectories of fire regimes and the impact of changing fire activity on the functioning of boreal ecosystems requires approaches with a broad spatial and temporal scope (Kasischke et al., 2010). Tree-dominated ecosystems have long successional cycles ranging between decades and centuries. Palaeoecological records capture long time scales and are, therefore, particularly suitable to track the fire dynamics of forest ecosystems (Whitlock et al., 2018). However, studies of past millennial-scale variability in fire regimes in Siberian boreal forests, forested peatlands and forest steppe started to emerge only recently (Feurdean et al., 2020a; Rudaya et al., 2020; Glückner et al., 2021; Barhoumi et al.,

2021; Novenko et al., 2022). Although Siberia is covered by extensive forests and forested peatlands, particularly to the west (Vompersky et al., 1994; Liss et al. 2001; Kirpotin et al., 2021), no studies have explicitly explored the interactions between fire regime, peatland moisture, and vegetation composition in this vast region.

In this study, we used multi-proxy analyses (pollen, non-pollen palynomorphs, microcharcoal, macrocharcoal morphologies, testate amoeba, geochemistry) on two new peat records from Tomsk Oblast to explore the patterns and drivers of the western Siberian fire regime throughout the Holocene. Specifically, we have explored how hydrological conditions in peatlands affect fuel dryness and how peat hydrology interacts with tree community composition with regard to plant functional types and fuel flammability in driving the frequency and type of fires. Combining our new data with our two other published records from Plotnikovo Mire (Feurdean et al., 2020a) allowed a quantification of the regional changes in biomass burning and fire frequency.

## 2 Material and Methods

### 2.1 Geographical location and site selection

The study areas stretch along a west-east transect in a permafrost-free area in Tomsk Oblast, in the southern taiga of western Siberia (Bleuten and Filippov, 2008; Kirpotin et al., 2021). The region has a continental boreal climate. The mean annual temperature is 0°C, the mean temperature of the coldest/warmest season is −17°C/17°C and mean annual precipitation is ~ 525 mm falling predominantly in the warm season (www.climexp.knmi.nl). Regionally, the forests are composed of both light (*Pinus sylvestris, Betula pubescens, B. pendula, Populus tremula*) and dark (*Pinus sibirica, Picea obovata,* and *Abies sibirica*) taiga species, the latter in greater proportions towards the east (Laschinsky and Koroliuk, 2014). In the southern taiga of Western Siberia, forests occupy 50% of the area, and peatlands 30% with an equal proportion of forested and open peatlands (Vompersky et al., 1994; Liss et al., 2001). Our coring sites are in open peatlands surrounded by forested peatlands within a 2-6 km radius, depending on the site (Fig. 1). Fires, at present primarily ignited by humans (80%), occur predominantly between spring and early autumn, with summer and autumn fires usually being the most severe (Kukavskaya et al., 2014, 2016; Kharuk et al., 2021).

We cored two sites, Rybnaya Mire (N57.27566800°, E84.48758900°) located near the Rybnaya river, a tributary of the Ob river, and Ulukh-Chayakh Mire (N57.34326197°, E88.32306189°) located on a terrace of the Chulym river, near the village of Teguldet, about 30-40 km from the border of the Krasnoyarsk Krai (Fig. 1). The local peatland vegetation at both coring sites includes mesotrophic open sedge-*Sphagnum* communities with small birch and pine trees at Rybnaya Mire and standing dead *Pinus* tree trunks at Ulukh-Chayakh Mire. These dead trees reflect a rise in water levels across the mire due to road construction in the 1960s. Peat cores (50 cm length, 5 cm diameter) totalling 400 cm in length at Rybnaya Mire and 348 cm at Ulukh-Chayakh Mire were extracted with a Russian-type corer in 2017 and 2019, respectively.

### 2.2 Chronology

We established the chronology of both cores based on AMS radiocarbon measurements performed at Isotoptech, Debrecen, Hungary (File S1). The [14]C AMS age estimates were converted to calendar years BP using the IntCal20 data set of Reimer et
al. (2020) and we constructed the age-depth models using the smooth spline method implemented in CLAM software (Blaauw, 2010). In the age-depth models, we assigned a surface age of -67 cal yr BP (coring year 2017 for Rybnaya) and -69 cal yr BP (coring year 2019 for Ulukh-Chayakh) to the surface samples of each sequence.

### 2.3 Fire history

**2.3.1 Charcoal-based reconstructions of fire history**

We inferred changes in the local-scale fire regime based on charcoal particles identified in peat samples of 2 $cm^3$ extracted at 1 cm contiguous intervals (Whitlock and Larsen, 2001). Preparation, identification and the categorisation of charcoal morphotypes followed Feurdean et al. (2020a) and Feurdean (2021). To deduce the predominant fire type, we pooled the charcoal morphologies into two categories: non-woody (graminoids, forbs) and woody types (deciduous leaves, wood, and resins) and
calculated their ratio. These two categories represent the dominant fuel sources for surface and crown fires, respectively (Enache et al., 2006; Jensen et al., 2007; Courtney-Mustaphi and Pisaric, 2014; Feurdean et al., 2017; 2020a; Feurdean, 2021). At Ulukh-Chayakh, we additionally tested the aspect ratio (length:width) of charcoal particles in selected samples to discriminate between the relative dominance of graminoids versus leaves and wood morphologies (Feurdean, 2021; Vachula et al., 2021). We grouped the charcoal fractions into two size classes (150-500 μm and >500 μm) to deduce the approximative
proximity of fires, i.e., local versus on-site fire events (Adolf et al., 2018). We calculated the influx of charcoal morphotypes and size classes (particles $cm^{-2}$ $yr^{-1}$) by dividing their respective concentration (particles/$cm^3$) by the sediment deposition rate determined from the age-depth models (yr/cm).

We estimated the frequency and severity of fire episodes using charcoal peaks identified from macrocharcoal particles using the method of Higuera et al. (2009). This methodology involved interpolating CHAR values to constant time steps (30 yr),
decomposing the record into a background component and a peak component (local fire episodes) using a robust LOWESS smoother (900 yr), and evaluating charcoal peaks using the 95th percentile of the modelled noise distribution. We used the signal-to-noise index (SNI) to assess the suitability of the record for peak detection (Kelly et al. 2011; File S2).

To integrate fire history across the study area, we created composite records of CHAR and fire frequency (FF) by combining the two new records with our two published profiles from Plotnikovo Mire (Feurdean et al., 2019; 2020a; Fig. 1). Composite
CHAR was created using the Power et al. (2010) protocol implemented in the R palaeofire package version 4.0 (Blarquez et al., 2014) with the following setup for Z score transformation: 8.5 ka to present as the base period and LOWESS smoothing to a 25-yr mean. The composite FF was created by pooling the identified fire events from all four records and applying a 150-yr smoothing.

### 2.3.2 Fire type identification from satellite images and forest statistics

To evaluate the ability of charcoal records to capture the occurrence and severity of fires near the two sites, we used satellite imagery of vegetation cover from LANDSAT 4-5 TM, LANDSAT-7 ETM+ and LANDSAT 8 OLI/TIRS (https://earthexplorer.usgs.gov/). We also used observation-based fire data from the Forest Department of the Tomsk Region from 1950 to 2020 CE and data from MODIS sensors onboard the NASA Terra-1 and Aqua-1, which detect forest fires (http://fires.ru/). All maps and images were processed using ESRI ArcGIS version 10.8.

### 2.4 Pollen-based reconstruction of vegetation dynamics

To reconstruct past vegetation dynamics, we analysed pollen and non-pollen palynomorphs (NPPs) mainly at 4 cm intervals in both profiles following the protocol of Bennett and Willis (2001). A pollen sum exceeding 500 terrestrial pollen grains, excluding Cyperaceae, was counted for most samples and converted into percentages based on the total terrestrial pollen sum. The percentages of Cyperaceae, ferns and mosses were calculated relative to the terrestrial pollen sum and their respective individual sum. The bottom part of the Ulukh-Chayakh profile, i.e., from 300-348 cm (age >6500 ka BP) contained extremely poorly preserved pollen and was excluded from further analysis. Microscopic charcoal particles longer than 10 μm were also counted alongside pollen, and their influx was estimated relative to the added *Lycopodium* marker and the sediment deposition rate.

We assigned pollen-derived tree taxa to one of the fire-related functional PFT groups - resisters (*Pinus sylvestris, Larix* spp.), avoiders (*Picea obovata, Abies sibirica, Pinus sibirica*), or invaders (*Betula* spp.). The two tree birch species (*B. pubescens* and *B. pendula*) and the dwarf shrub (*B. nana*) could not be differentiated by pollen analysis and were considered as *Betula* spp. However, at present day, *B. nana* tends to be found only sparsely in this region. Due to its poor pollen preservation, *Populus,* a fire endurer*,* is mostly absent from our pollen records. The necessity to combine the *Betula* pollen signal and the paucity of pollen of *Populus* may have resulted in the endurer PFT group failing to be represented in our records. The PFT-specific fire coping adaptations relate to a taxon's strategy to complete its life cycle in the context of a given fire regime (Gill, 1981; Wirth, 2005). We also separately grouped the tree pollen into taxa associated with dark taiga (*Pinus sibirica, Abies sibirica, Picea obovata*) and light taiga (*Pinus sylvestris, Betula* spp*., Larix* spp.) and created an index for dark-to-light taiga composition, an approximation of forest density, by calculating the ratio between the relative abundance of dark and light taiga tree taxa (Higuera et al., 2014). Additionally, we pooled the non-arboreal pollen types as shrubs, herbs, ferns and mosses. To determine the regional changes in tree community composition, we created composite records of PFTs. This was done by averaging the Z-scores and smoothing them to 300 years with a locally weighted regression and a 95% confidence envelope on the same sites used for composite biomass burning reconstructions in the R palaeofire package.

### 2.5 Climate reconstruction based on testate amoebae and water table depth

To determine changes in peatland hydrology at each site, we used testate amoeba-based water table reconstructions. The testate amoebae preparation and identification were conducted at 4 cm intervals following the methods of Hendon and Charman

(1997) and Charman et al. (2000). Based on the available literature (Grospietsch, 1958; Ogden and Hedley, 1980; Mazei and Tsyganov, 2006), we identified and counted a minimum of 150 testate amoebae per sample. No testate amoebae were present in core segments below 285 cm (age > 6000 ka BP) in the Ulukh-Chayakh sequence. To derive the water table depth, we used two transfer functions, one developed for the pan-European region (Amesbury et al., 2016) and the other for Asia (Qin et al., 2021). The sample errors were generated using 1000 bootstrapping cycles (Line et al., 1994). The two transfer functions applied to our testate amoebae records show similar trends but different absolute water-table depth values (cm). To avoid potential misinterpretation of absolute water table depth values, we followed literature recommendations (Amesbury et al., 2016; Swindles et al., 2016; 2019; Qin et al., 2021) and standardised the cm values and presented them as residual values (Z score). For the composite water table reconstruction, we used the same setting as for pollen.

**2.6 Elemental geochemistry**

The concentration of the geochemical element Ti was measured using a non-destructive Niton XL3t 900 X-Ray Fluorescence analyser (fpXRF) to determine the potential influence of water influx (i.e., floods) on mire water table. Sedimentary Ti NCS DC73308 was used as a Certified Reference Material (CRM). Measurements followed the procedure described by Hutchinson et al. (2016).

**2.7 Numerical methods**

To quantify the relationship between peatland moisture conditions, biomass burning, the relative abundances of PFTs and dark-to-light taiga composition, we performed a correlation analysis. We used microcharcoal (CHAR particles <150 μm) and machrocharcoal (CHAR particles >150 μm) influx to determine regional and local biomass burning, pollen percentages as indicators of PFTs (resisters, invaders, avoiders, and others) at local to regional scales, and testate amoebae-based water table depth as a proxy of local peatland moisture. The group "others" includes pollen and spores of shrubs, herbs, ferns and mosses. Before the analysis, we interpolated all datasets fed into the model to a 100- year interval using linear interpolation.

**3. Results**

**3.1 Chronology**

The age-depth model of the Rybnaya sequence spans ~ the last 8400 years and shows a mean peat accumulation rate of 25 yr/cm (ranging between 6-36 yr/cm). The Ulukh-Chayakh sequence may cover ~ the last 8500 years, but the chronology of the bottom part of this site (>6000 years) relies on linear extrapolation, therefore is highly uncertain (File S1). The subsequence covering the last 6000 years has an average temporal resolution of 21 yr/cm (ranging between 3-37 yr/cm).

**3.2 Fire type identification from charcoal, satellite images and forest statistics**

The comparison of sedimentary charcoal from sub-recent samples with satellite imagery of the Plotnikovo Mire (sites SP and SD) confirmed that the marked increase in charcoal particles towards the present day at the SD location closely agrees with

the occurrence of high-severity fires (Fig. 1; Feurdean et al., 2020a). Satellite images and official forest statistics show no evidence of recent fires in the vicinity of the Rybnaya and Ulukh-Chayakh sites, except for a small fire documented in 1993 near the Ulukh-Chayakh site (Fig.1). This aligns with the scarcity of charcoal pieces from sub-surface samples in both cores (Fig. 2; File S3). A high-severity fire event occurred in 2015 some ca. 30 km from Ulukh-Chayakh, which is not documented in our sub-surface charcoal record and corroborates the localised origin of charcoal found in this sedimentary profile. Samples from the Ulukh-Chayakh site show a minor increase in charcoal (2-7 pieces; up to 0.6 cm$^{-2}$ yr$^{-1}$) ca. fifty years ago, possibly related to a local fire event in 1993. The first clear charcoal peak at this site (250 pieces; 7.3 cm$^{-2}$ yr$^{-1}$), with numerous wood fragments, was found ca. 200 years ago, whereas the preceding charcoal peak (400 pieces; 9.2 cm$^{-2}$ yr$^{-1}$) was ca. 900 years ago (Fig. 2; File S3). These two peaks therefore indicate the occurrence of infrequent, high-severity local fires that produce large quantities of charcoal, embedded in what is otherwise predominately a surface fire generated low level of charcoal.

### 3.3 Site-specific and composite record of biomass burning and fire frequency

Both records displayed a high signal-to-noise index (File S2) with almost all peaks well above 3 (a mean of 4.98 at Rybnaya and 6.45 at Ulukh-Chayakh), the theoretical minimum value for justification of peak analysis (Kelly et al., 2011). At Rybnaya, CHAR was highest between 7.5 and 6 ka BP and around 4.5 ka BP, whereas the frequency of high-severity fires inferred from the charcoal peak magnitude and abundance of woody morphologies ranged between 1 and 2 fires/900 yr (Figs. 2, 4; Files S2, S3). At Ulukh-Chayakh, CHAR was highest >6 and 3.5 ka BP, and the fire frequency varied between 2 and 5 fires/900 yr. Both profiles showed a decline towards a minimum in CHAR and frequency of severe fires, i.e., low peak magnitude and reduced occurrence of woody charcoal morphologies, approximately between 4 and 1.5 ka BP (3-1.5 ka BP at Ulukh-Chayakh). The CHAR increased towards the present at both sites, but the charcoal peak amplitude was considerably higher at Ulukh-Chayakh.

The composite CHAR record showed augmented regional biomass burning between 7.5 and 4 ka BP (positive Z-scores) with peaks centred at 7.5 and 4.5 ka BP (Fig. 5). We found a pattern of CHAR decline towards a minimum in the composite profiles between 4 and 2 ka BP (negative Z-scores), followed by another increase between 2 ka BP to the present (positive Z-scores). The composite fire frequency showed an increase until ca. 5-4.5 ka BP and then again from 2 ka to the present.

### 3.4 Site-specific and composite record of vegetation composition and forest density

Early Holocene data is only available from the Rybnaya site (8.5 and 7.5 ka BP), indicating that light taiga and the fire-invader *Betula* were most abundant (up to 60%) during that period, whereas other light taiga fire-resisters (*Pinus sylvestris)* and dark taiga and fire avoiders (mainly *Picea obovata),* had an abundance of 10% and 5%, respectively (Fig. 3; File S4a). The high proportion of light taiga resulted in a low dark-to-light taiga ratio of 0.1, and perhaps a low forest density. An increased dark-to-light taiga ratio (> 0.2), was identified between 7.5 and 4 ka BP at Rybnaya and between 5 and 2.5 ka BP at Ulukh-Chayakh (Fig. 5; File S4a,b). The light taiga was composed of up to 70% fire-resisters (*Pinus sylvestris* and *Larix spp.),* while the dark taiga, with its fire avoiders (*Pinus sibirica, Picea obovata,* and *Abies sibirica*), made up to 25% of the forest. A decline in fire-

resisters (down to 50%), avoiders (below 10%) and dark-to-light taiga ratio and increasing invader abundance (up to 60%) occurred after 4.5 ka BP at Rybaya and after 2.5 ka BP at Ulukh-Chayakh. Between-site variability was highest within the avoider group. *Picea obovata* had higher values at Rybnaya (up to 10%) than at Ulukh-Chayakh (3%), whereas *Abies sibirica* was more abundant at Ulukh-Chayakh (15%) than at Rybnaya (7%; Fig. 4). The composite vegetation record showed a higher proportion of fire-resisters between 7.5 and 5 ka BP, fire avoiders between 7.5 and 3 ka BP, and fire invaders over the last 4.5 ka BP (Fig. 5).

Understory vegetation primarily reflects local peatland development. It was predominantly composed of herbs/sedges (Cyperaceae; up to 20%) and ferns (Polypodiaceae, up to 20%) and *Equisetum* (up to 25%) at both sites until ca. 4.5 ka BP. An increase in the abundance of shrubs (Ericaceae; up to 10%) and moss (*Sphagnum;* highly fluctuating between 10 and 100%) at Rybnaya, and Ericaceae (up to 5%) and moss (*Sphagnum;* up to 100%) at Ulukh-Chayakh was noted over the last 4.5 ka BP (Fig. 3; File S4a,b). The sum and individual levels of herbaceous taxa, including those potentially related to human impact (*Artemisia*, Asteraceae, Chenopodiaceae, Urticaceae, Brassicaceae, *Plantago lanceolata*), were constant throughout the profiles. However, their abundance increased slightly over the last 1.5 ka BP (Fig. 3; File S4a,b). Coprophilous dung spores, *Podospora* and *Sporormiela,* were present throughout both records, although they became more abundant between 4.5 and 3.5 ka BP and 3.2 and 1.0 ka BP at Ulukh-Chayakh and over the past millennium at Rybnaya (File S4a,b).

### 3.5 Site-specific and composite records of peatland moisture conditions

The two testate amoebae transfer functions returned similar trends for the water table depth, but the absolute values (cm) were different; the Asian transfer function yielded higher reconstructed water tables than the European transfer function (Fig. 4; File S5). At Rybnaya, the standardised values of water table depth indicated relatively high levels (negative Z score) between 8.5 and 7.5 ka BP, 4 and 2.5 ka BP, and over the last 1 ka BP (Fig. 4). We reconstructed low water levels (positive Z score) for the time interval between 7.5 and 4.5 ka BP and 2.5 and 1 ka BP (Fig. 4). At Ulukh-Chayakh, the water level was high (negative Z score) between 3.5 and 2.5 ka BP, 2 and 0.5 ka BP, and towards the present day. There, times of low water levels (positive Z score) occurred between 6 and 3.5 ka BP, 2.5 and 2 ka BP, and around 0.5 ka BP. The composite record of the water table showed high levels approximately between 8.5 to 7.5 ka BP, 3.5 and 3 ka BP and during the last 1.5 ka BP, and low levels between 7.5 and 4.5 ka BP, and around 2.5 ka BP (Fig. 5).

### 3.6  Numerical analysis

The correlation analysis at Rybnaya showed that micro- and macrocharcoal were highly correlated and positively associated with the proportion of fire avoiders and light-to-dark taiga index and negatively with resisters and invaders (Fig. 6a). However, not all these relationships were statically significant (Appendix A1). At Ulukh-Chayakh, micro- and macrocharcoal were weakly correlated. Microcharcoal was positively associated with fire avoiders, resisters, others and light-to-dark taiga index and negatively with invaders (mostly statically significant; Fig. 6a, b). All other relationships i.e., between PFTs and macrocharcoal and depth to water table were statistically non-significant (Fig. 6a; Appendix A1).

## 4 Discussion

### 4.1 Changes in fuel, fire type and fire frequency over the Holocene

Stand-replacing fires combust substantial amounts of biomass because they burn entire trees. Such fires manifest themselves in a greater abundance of charcoal particles that exceed the quantities typically produced by surface fires (van Marle et al., 2017). This feature allows a distinct separation of charcoal peaks from charcoal background levels, which yields a higher signal-to-noise index (Higuera et al., 2005; 2009; Courtney Mustaphi and Pisaric 2014). The comparison of sedimentary charcoal from sub-recent samples with satellite imagery of the Plotnikovo Mire (Feurdean et al., 2020a), Rybnaya and Ulukh-Chayakh show that high-severity local fires produce large quantities of charcoal; thus the charcoal peak extraction method applied here reliably reflects the occurrence of such fires.

We reason that the prevalence of a high charcoal influx with well-defined charcoal peaks and abundant woody morphotypes found between ca. 7.5 and 4 (3) ka BP at both sites, and over the past 1.5 ka BP at Ulukh-Chayakh, is indicative of the predominance of high-severity local fires (Fig. 2; File S3). Between 4 (3) and 1.5 ka BP, the observed low charcoal influx and peak magnitude were coupled with a lower abundance of woody morphologies at both sites, although the latter feature was more pronounced at Ulukh-Chayakh (Fig. 2; File S3). For this period, we inferred predominately low-temperature surface fires that mostly burned understory biomass. Experimental production of charcoal additionally provides evidence that graminoids and *Sphagnum* retain a high charred mass only during low-temperature fires and that their charred mass declined rapidly with increasing fire temperature (Hudspith et al., 2017; Feurdean, 2021). Measurements of aspect ratio ($L{:}W$) and surface area on selected charcoal samples at Ulukh-Chayakh show that the aspect ratio only partially agrees with the predominant fuel type inferred from charcoal morphologies. A lower aspect ratio of charcoal particles, typical for wood and leaf morphotypes, was what we expected at times of increased woody morphologies, and an increased aspect ratio with a rise in the relative proportion of graminoids (Feurdean, 2021; Vachula et al., 2021). This is probably the result of a strongly mixed morphology of charcoal types in the same sample (Fig. 2). Our measurements of charcoal surface area indicate high values at times of an increased abundance of leaves (Fig. 2; File S3) with a typically high surface area (Crawford and Belcher, 2014; Feurdean, 2021).

The reconstructed fire return interval is ~ 450 yr (2 fires/900 yr) for both sites throughout most of the Holocene (Fig. 4). However, fire frequency (at Rybnaya) and severity (at Ulukh-Chayakh) increased over the last 1.5-2 ka BP compared to the long-term averages of both records suggesting an intensification of fire activity during the most recent two millennia. The composite fire frequency derived from all four records also shows increasing biomass burning and fire frequency over the last 2-1.5 ka BP (Fig. 5). The reconstructed fire return interval (FRI) in the study area is largely in line with other sites located in forested peatlands (*Pinus-Betula* dominated, *Picea obovata, Abies sibirica*) in Russia, which report an FRI range between 100 and 600 yr (Barhoumi et al., 2019, 2020; 2021). In contrast to a study in the northern Ural region (Barhoumi et al., 2019), we have not found a gradual increase in fire frequency from the early Holocene towards the present, but rather two distinct periods of enhanced activity, between 7.5 and 4 ka BP, and approximately over the last 2 ka BP (Fig. 5), a pattern more similar to sites in the West Siberian Plain (Rudaya et al., 2020) and Lake Baikal area (Barhoumi et al., 2021).

### 4.2 Drivers of vegetation and fire regime change

Climate conditions influence peatland hydrology, fuel dryness and flammability, as well as vegetation composition with regard to moisture- and fire-related PFTs. The relative abundances of the fire-related PFTs, in turn, influence the prevalence of specific fire regime types, e.g., low-intensity vs. high-intensity regimes. Below, we discuss how hydrological conditions in peatlands affect fuel dryness and flammability and feedback between tree community composition (PFTs) and fire regime.

### 4.2.1 The influence of peatland moisture on fuel type and flammability

Waterlogging in peatlands and the occurrence of a dense *Sphagnum* cushion can limit fire severity and reduce the depth of the burnt horizon. Moreover, a waterlogged horizon provides a substrate that serves as a seedbed for post-fire regeneration (Whitman et al., 2018; Gewin 2020). However, increasingly dry peatland conditions and consequently greater tree cover can reduce the limiting effect of waterlogging on fire severity (Magnan et al., 2012; Kettridge et al., 2015; Whitman et al., 2018; 2019;

Loisel et al., 2020). Deep burns can even smoulder over winter and re-ignite the following spring (Scholten et al., 2021). We found that the probability of fire occurrence and the intensification of fire severity was greater at times of low water level (drier peatland conditions) and decreased at high water levels (wetter peatland conditions; Figs. 4, 6). Wet peat surface conditions likely prevented the fire from spreading on the peatland and in the surrounding forests, but also the propagation of fires from the surrounding forests across the peatland. The stronger association between peatland moisture conditions and burning at

Rybnaya may be related to the higher amplitude of changes in local hydrological conditions and the overall drier conditions at this site compared to Ulukh-Chayakh (Fig. 4; File S5).

Water levels of mesotrophic mires are co-regulated by the balance between precipitation (P) and evapotranspiration (Et), and external water inflow (Chambers and Charman, 2004). The detrital element Ti, a possible indicator of water influx in minerogenic mires, increased notably between 4 and 3 ka BP and 1.5 and 0.1 ka BP at Ulukh-Chayakh, but only slightly over the last

1 ka at Rybnaya (File S5). This suggests increasing fluvial input, reflecting possible flooding or channel position change as a transport mechanism for the delivery of such material at Ulukh-Chayakh (File S5; Leshchinskiy et al., 2011). Mire type development and changes in peat plant communities could have also influenced peatland hydrology (Galka et al., 2016; Kurina and Li, 2019; Blyakharchuk and Kurina, 2021). At Rybnaya, the minerogenic stage of the mire occurred from 8 to 4.5 ka BP and the water level was more stable. With the meso-oligotrophisation of the mire, indicated by the appearance of Ericaceae at 4.5

360   ka BP, and the dominance of *Sphagnum* from 3.6 ka BP to the present, the water level and the amplitude of its fluctuation rose and the occurrence of peat fires declined (Figs. 3, 4; File S4a,b). At the Ulukh-Chayakh site, Ericaceae developed at 4.5 ka BP and remained dominant up to the present, whereas *Sphagnum* established around 2 ka BP and became dominant only in the past decades (Fig. S4). Further, mire margins with thinner peat depth burn more frequently than thicker peat (Turunen et al., 2001). Thus, the location of the Ulukh-Chayakh sampling point close to the mineral margin of the mire (ca. 160 m), as opposed

to a few kilometres at Rybnaya, could have sustained a high fire occurrence at this site until recently. Lastly, fire itself can

influence peatland hydrology, with field measurements indicating an age-dependent water table lowering after burning (Holden et al., 2015).

At a Holocene scale, the intensity of fires and/or fire size was greatest between 7.5 and 6.5 ka BP and 5 and 4 ka BP, but the timing was not fully synchronous between the sites (Figs. 4, 5). From 9 to 4.5 ka BP, annual temperatures in the Northern Hemisphere were up to 3.5°C warmer than at present (Fig. 5) and exhibited a more pronounced seasonality, i.e., higher summer and lower winter temperatures (Kauffman et al., 2020; Bova et al., 2021). Pollen-based climate reconstructions in south Siberia show warmer and drier-than-present climate conditions in the early Holocene (until 8 ka BP), a continuous increase in moisture conditions to the present, and a temperature and moisture optimum approximately between 6 and 4.5 ka BP (Borisova et al., 2011; Groisman et al., 2012; Zhang and Feng, 2018). A review of multiproxy palaeohydrological conditions along a west to the east gradient in southern Siberia shows strikingly similar hydrological conditions to ours for the western part, i.e., moist between ca. 8.1 and 7.4 ka BP, followed by dry conditions between ca. 7.4 and 5.0 ka BP, associated with a diminished North Atlantic influence (Mikhailova et al., 2021). Testate amoebae-based hydrological reconstructions in the southern taiga of West Siberia, however, show considerable temporal variability in peat moisture conditions between sites, and only a coeval shift to dry conditions between 7.0 and 6.0 ka BP (Kurina et al., 2018), primarily linked to declining precipitation.

The interval of lowest biomass burning and fire severity approximately between 4 (3) and 1.5 ka BP coincides with one of the wettest peatland conditions during the Holocene at both sites (4.5-2.5 ka BP) followed by mostly dry peatland conditions (Fig. 4). Annual and summer temperatures declined after 4.5 ka BP in the Northern Hemisphere (Kauffman et al., 2020; Bova et al., 2021), and moisture was high for most of the past 4.5 ka BP in the southern part of western Siberia (Zhang and Feng, 2018; Mikhailova et al., 2021). Testate amoebae-based hydrological reconstructions in the southern taiga of western Siberia, however, show long periods with dry conditions over the past 4.5 ka, with short-term wet spells occurring at 2.6-2.3 ka BP, 1.3-1.1 ka BP, 0.9-0.2 ka (Kurina et al., 2018) or over the past 2.5 ka (Blyakharchuk and Kurina, 2021). Although biomass burning and the frequency of severe fires show more centennial-scale variability over the past 1.5 ka BP, this matches only partially known climatic fluctuations, such as the Medieval Climate Anomaly (MCA 1.2-0.65 ka BP) and the Little Ice Age (LIA; 0.7-0.4 ka BP). Notably, an increased charcoal peak amplitude occurred during the second part of the LIA (0.4-0.2 ka BP), at a time of extremely wet conditions (Fig.4; Feurdean et al., 2019). We hypothesise that the climatic linkage between the cool and wet conditions, that implies high fuel moisture conditions and typically low fire activity, was likely disrupted by anthropogenic activity towards the present day (Fig 3; File S4a, b), as document by other pollen records in the region (Blyakharchuk et al., 2019). Historical records indicate the widespread colonisation of the region by Russian (i.e., from the west) at this time, associated with the conversion of forest to arable land and pastures, and likely increased anthropogenic ignitions (Naumov, 2006). Most recently, local fire suppression near the village of Teguldet (near Ulukh-Chayakh mire) may have hampered contemporary fire spread on the mire leading to the recent pattern of low fire activity.

Nevertheless, humans may have altered the frequency of fires earlier than the widespread Russian colonisation of the region. A GIS survey of archaeological finds in the Tomsk region reveals a high density of sites, but no distinctions between the

cultural phases (Zolnikov et al., 2020). In parallel with modern settlements distribution, most of these ancient sites were situated near rivers favouring the floodplain terraces of river valleys (Zolnikov et al., 2020). Another study in the Lower Tom basin attests to the presence of tools that have the characteristics of sites commonly associated with the Neolithic and Bronze Age close to Rybnaya (Idimishev et al., 2018). There is little evidence of pollen indicative of human impact at Rybnaya until 400 years ago, but coprophilous-based evidence of herbivores (*Podospora*) is present from 7 ka onward and increased over the past 400 years, whereas fire activity was high from 7 to 5 ka BP (Fig. 4, S4). At the Ulukh-Chayakh site, coprophilous spores became more abundant between 4.5-3.5 ka BP (*Sporormiella)* and 3.2 and 1.0 ka BP (*Podospora* and *Sporormiella*) while fire activity increased between 4 and 3 ka and over the last millennium. However, it is difficult to say from our records whether the coprophilous remains originated from wild or domesticated animals and how much herbivores contributed to the opening of the forest and facilitated the fire spread (Morales-Molino et al., 2019). Given the locations of our sites near rivers, the archaeological evidence suggests the possibility of some level of human impact on vegetation and fire.

### 4.3 Feedbacks between the PFT composition of vegetation and fire regime

Climate conditions affect the relative abundances of PFTs. Fire-related PFTs in turn, create positive or negative feedbacks for fire regimes via specific fire-related traits (Feurdean et al., 2020a; Kharuk et al., 2021). We found a dominance of open light taiga and fire invaders species (*Betula* spp) between 8.5 and 7.5 ka BP and over the past 4.5 ka BP. The occurrence of *Betula*-dominated forests coincides with marked variability in the frequency and severity of fire, i.e., moderate between 8.5-7.5 ka BP, low between 4.5 and 1.5 ka BP, and high over the past 1.5 ka BP (Figs. 2, 4). *B. pubescens* grows on wet peatlands, whereas *B. pendula* grows more frequently on drier sandy soils and peatlands with higher fire incidence (Wirth, 2005; Groisman et al., 2012; Blyakharchuk, 2003). The prevalence of birch species with contrasting hydrological and fire requirements (unable to be differentiated by pollen) may explain such variable association of *Betula* and fire regime. Denser forests of light taiga and fire-resisters (*Pinus sylvestris*), interspaced with dark taiga and fire avoiders, primarily *Pinus sibirica* at Rybnaya and *Abies sibirica* at Ulukh Chayakh (only from 5.5 ka BP), prevailed between 7.5 and 4.5 ka BP (Fig.3; File S4a,b). This forest composition coincides with increased fire severity. Visual trends and correlation reveal that the biomass burning increase at intermediate dark-to-light taiga index (ranging from 0.15 to 0.3; Figs. 4, 6). This is consistent with emerging findings pointing to a higher fire hazard in less dense as opposed to denser boreal forests, which allow fuel to dry (Scheffer et al., 2012; Feurdean et al., 2020b). The dominance of dark taiga forests in western Siberia was associated with increased moisture conditions (Blyakharchuk and Sulerzhitsky, 1999; Zhang and Feng, 2018; Barhoumi et al., 2021). Our correlations, however, suggest a greater tolerance of deciduous trees to shallow water level (wetter conditions) compared to conifers (Fig. 6 a,b). Note that testate amoebae and macrocharcoal reveal local changes in peatland hydrology and fire activity, whereas pollen and microcharcoal reflect local to regional changes in vegetation composition (the tree composition on the peatlands and adjacent forests) and fire activity (Bennet and Willis, 2001; Whitlock and Larsen, 2001; Lamentowicz et al., 2015; Blyakharchuk and Kurina, 2021). The deviating scale-dependency of the reconstructions may induce some uncertainty in linking the local peat moisture conditions and the dominant local to regional PFTs.

The presence of the fire-resisters (i.e., *Pinus sylvestris*) in Siberia is associated with a light surface fire regime (mean FRI of 28 years) that is occasionally interrupted by longer-term (mean FRI of 200 yr) stand-replacing crown fire events (Ivanova, 2005; Kukavskaya et al., 2016). Fire avoiders typically experience severe stand-replacing crown fires at relatively long return intervals (mean FRI 150 years; range 99 and 300 years; Goldammer and Furyaev, 1996; Wirth, 2005; Kharuk et al., 2021). The post-fire regeneration pathway of conifer species involves recruitment directly after burning (*P. sylvestris*) or after an early successional phase initiated by *Betula* (*Pinus sibirica, Picea obovata,* and *Abies sibirica*). A mixed taiga forest with deciduous trees and dark taiga conifer taxa typically develops between 60-120 years after a stand-replacing fire, followed by the dominance of conifers as a late-successional stage (Tautenhahn et al., 2016; Coop et al., 2020; Kharuk et al., 2021). We reason that the prevailing periodicity of severe fires (every 180 to 450 years) between 7.5 and 4.5 ka BP provided fire-free periods that were long enough for the fire avoiders *Pinus sibirica, Picea obovata* and *Abies sibirica* to reach reproductive age. Alternatively, burning may have been too patchy to create sufficiently large forest gaps to limit seed dispersal and inhibit post fire recovery. *Pinus sibirica* is typically classified as a fire avoider but behaves like a fire resister when reaching old age, which could also explain its increased presence at times of high fire activity. Fire also allows dark taiga species to compete with *P. sylvestris* and become established beyond poor soils and boggy areas (Kharuk et al., 2021).

A different fire regime and altered vegetation feedbacks emerged over the last two millennia at Ulukh-Chayakh, the site that experienced more severe fire episodes. The intensification of local fire severity at Ulukh-Chayakh paralleled a decline in the proportion of fire avoiders (mostly *Abies sibirica* and *Picea obovata*), an increase in fire invaders (*Betula)* and the abundant occurrence of heathland shrubs (Figs. 3. 4). Shrubs burn hotter than other surface fuel types and reach temperatures that kill most conifer seedlings (Tautenhahn et al., 2016). High-severity burned areas have also been found to be more prone to repeated burning and have a more negative impact on conifers (Kukavskaya et al., 2016; Whitmann et al., 2019; Coop et al., 2020). We propose that the intensification in fire severity at Ulukh-Chayakh may have eliminated mature *Abies sibirica* and *Picea obovata,* limited their seed dispersal and disrupted their successional pathways. These dynamics resulted in a shift in tree community composition towards more post-fire-adapted invader communities with better dispersal capabilities, recruitment strategies for burned areas and rapidly maturing taxa. Our results support emerging findings of increases in deciduous trees (*Betula* or *Populus*) at the expense of evergreen conifers that are associated with contemporary warming and increasing fire severity in boreal forests and forested peatlands in Siberia and North America (Kelly et al., 2013; Tautenhahn et al., 2016; Mekkonen et al., 2019; Whitman et al., 2019; Kharuk et al., 2021). At Rybnaya, the modest increase in biomass burning over the past two millennia coincided with wet peatland conditions, the occurrence of *Sphagnum*, and a forest composition that remained dominated by *Betula*, indicating that fire has played a more limited role in tree community dynamics at this site. The post-disturbance (fire, insect outbreak) forest regeneration pathway in Siberia also involves *Populus tremula.* Eurasian aspen forest communities can reach hundreds of years in age and change forest composition to *Populus* dominance where regeneration of dark taiga conifers is absent (Tautenhahn et al., 2016; Kharuk et al., 2021). Although we have not encountered *Populus* in our pollen record due to its poor pollen preservation, it cannot be discounted that *Populus* may have been a part of post fire forest succession pathways in the past.

**Conclusion**

This study provides novel insights into past fire regimes based on Holocene records from western Siberia. It demonstrates that peatland hydrology is a critical factor of fuel dryness, peat plant composition and fire-related plant functional types, and fire regime. We found that the probability of fire occurrence and intensification of fire frequency and severity increased at times of low water table (drier conditions), enhanced fuel dryness and flammability, and an intermediate dark-to-light-taiga ratio. Wet peatland conditions promoted deciduous trees typically classified as light-taiga taxa and fire invaders (*Betula*), whereas dry peatland conditions facilitated conifers and, therefore, a high dark-to-light-taiga forest ratio.

At a Holocene timescale, we found an enhanced fire severity between 7.5 and 4.5 ka BP associated with a higher proportion of dark taiga/fire avoider taxa (mainly *Pinus sibirica* at Rybnaya and *Abies sibirica* at Ulukh Chayakh) in a light taiga and fire resister (*Pinus sylvestris*) dominant community and drier peatland condition. The second period of increased fire severity (i.e., last ca. 1.5 ka BP) coincided with a reduced abundance of fire avoiders (mostly *Abies sibirica* and *Picea obovata*) and an expansion of fire invaders (*Betula*). These community changes demonstrate that frequent fires of higher severity can lead to compositional changes in forest tree communities, either because the trees were unable to reach their reproductive age between burning events, or the fires created substantial forest gaps that hindered seed dispersal. This study also shows certain prolonged periods of synchronous fire activity across the sites, suggesting that the magnitude of centennial to millennial-scale climate variability over the Holocene was marked enough to drive fire regimes on a regional scale.

Based on our findings from fossil records, the frequent warm and dry spells predicted by climate change scenarios in Siberia for the future (IPCC, 2021) will likely enhance peatland drying and may convey a competitive advantage to some conifers, especially *Pinus sylvestris*. But dry peatland conditions will also exacerbate the likelihood of greater fire frequency, increase fire severity (i.e., stand replacing fires) and likely disrupt the successional pathway of typical taiga conifers, notably that of fire avoiders. Such a fire regime change may accelerate a shift towards communities dominated by deciduous fire invader-type trees. Future climate-disturbance-fire feedbacks will not only accelerate changes in the boreal forest structure and composition, but also impact the carbon balance of the forested peatlands and ultimately lead to albedo-mediated feedbacks on the regional climate system.

**Appendices**

**Appendix A1.** Correlation coefficients for the Rybnaya and Ulukh-Chayakh sequences. Note that forest density refers to dark-to-light taiga ratio.

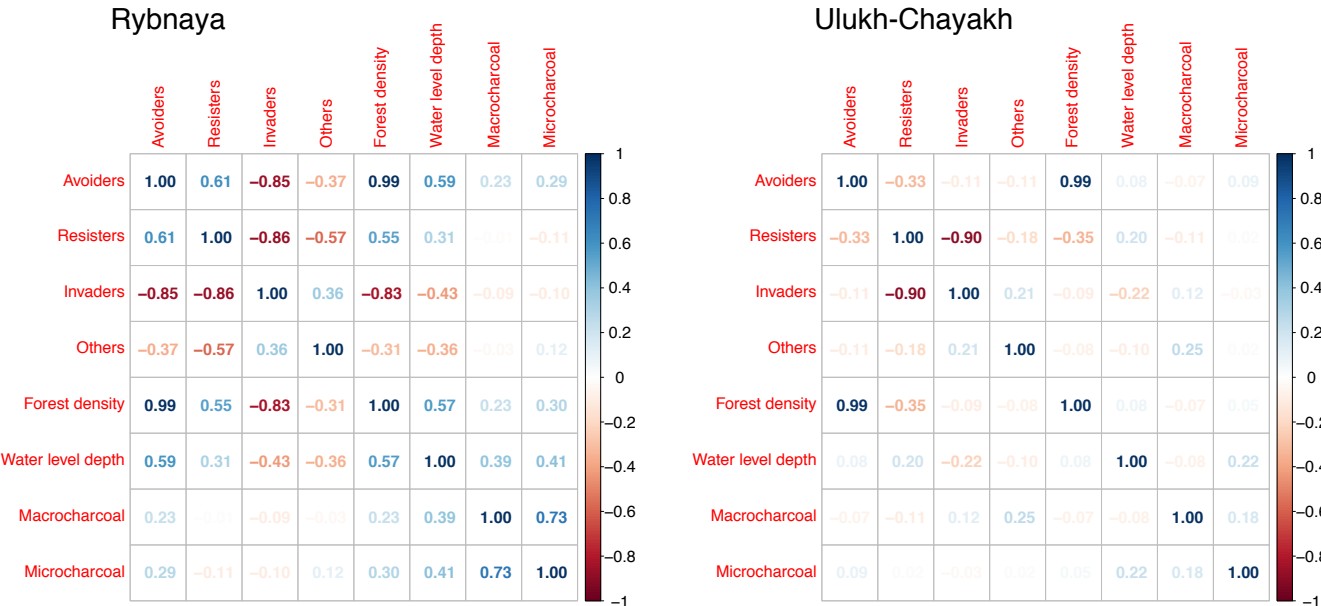


**Data availability**

All data generated for this paper will be deposited in the Neotoma database following publication.

**Author contribution**

Conceptualization: AF; Methodology: AF, ACD, MG, SMH, GB, NG, ST, IT, SK; Investigation and writing: AF prepared the manuscript with contribution from MP, and SMH; Editing: ACD, MP, MG, GB, NG, SMH, ST, AN, IO, SK; Funding acqui-510 sition: AF.

**Competing interests**: The authors declare that they have no conflict of interest.

**Acknowledgements**

We thank Rosa Rytkönen, University of Manchester for her help during fieldwork and the reviewers for useful comments to improve the manuscript. Sergey Loiko also provided archaeological literature from the region.

**Funding**

This research was supported by the German Research Foundation (grant no. FE-1096/6-1). SK is grateful for support from the
Russian Science Foundation (No. 20-67-46018) and the Tomsk State University Development Programme («Priority-2030»).

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

**Figures**

**Figure 1.** Location of the study area in Eurasia, Russia and the Tomsk region (A). Satellite based images showing the location and spatial extent of vegetation types and the fire event at the previously published sites on the Plotnikovo Mire (B; Feurdean et al., 2020a) and the new study sites: Rybnaya (C) and Ulukh-Chayakh (D).

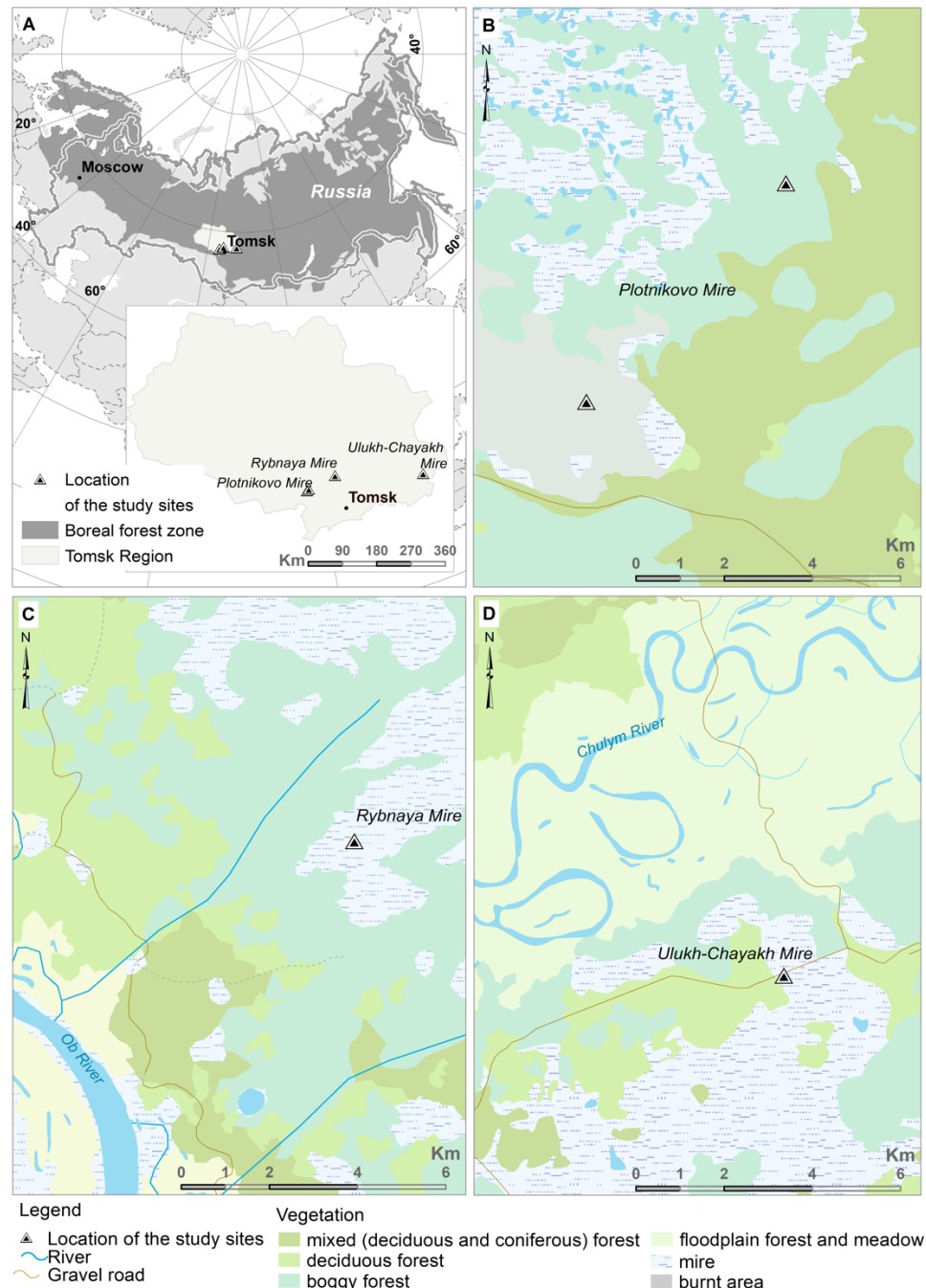

Legend

Location of the study sites
River
Gravel road

Vegetation

mixed (deciduous and coniferous) forest
deciduous forest
boggy forest

floodplain forest and meadow
mire
burnt area

**Figure 2.** The ratio of charcoal morphotype influx (#/ cm$^{-2}$ yr$^{-1}$) of the two main categories: woody (wood, deciduous leaves, resins) and non-woody type (forbs, grass) in the Rybnaya and Ulukh-Chayakh sequences. The aspect ratio (*L:W*) at Ulukh-Chayakh is used for fuel type identification. A high *L:W* ratio is typical for a higher abundance of graminoids charcoal, whereas

a lower *L:W* ratio is typical for charcoal from wood and leaves. The ratio of charcoal influx (#/cm$^{-2}$ yr$^{-1}$) of the two main size classes: small 150–500 μm and large > 500 μm in the two sequences and the charcoal surface area (μm$^2$) at Ulukh-Chayakh

(see File S3 for a full range of size classes). Bullets at Ulukh-Chayakh represent the extra-large charcoal fraction (>1000 μm) identified during routine plant macrofossils analysis of sediment volumes of ca. 20 cm$^3$. The grey time window highlights the period with an uncertain chronology.

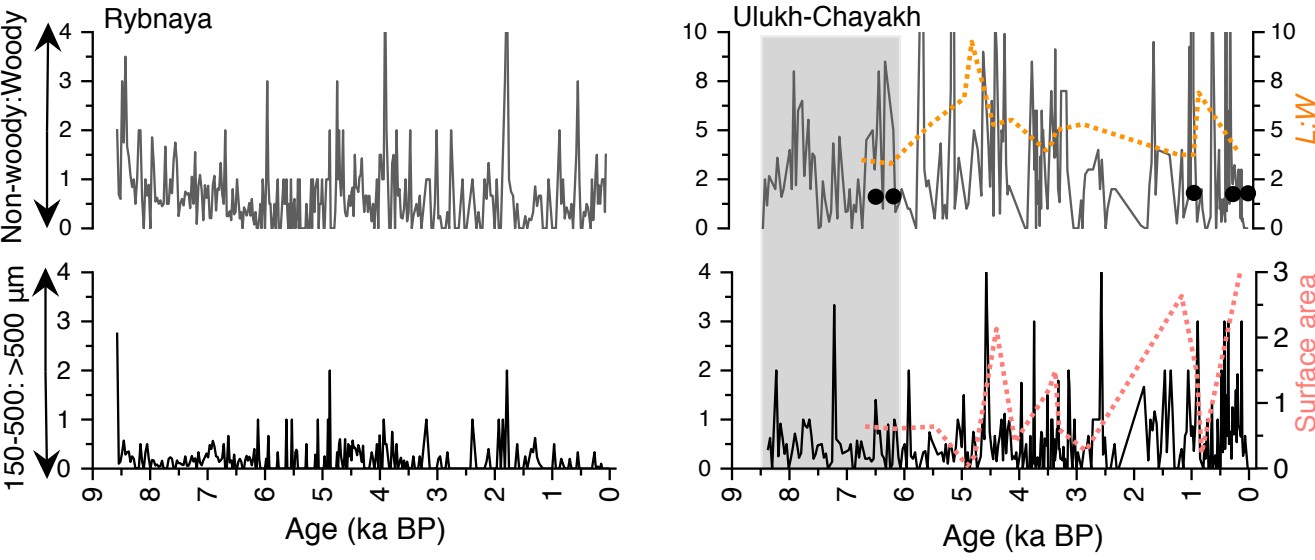




**Figure 3.** Temporal trends in the pollen and spore percentages of individual tree taxa and lumped group-wise for shrub, herb, fern and moss at Rybnaya and Ulukh-Chayakh. Periods with substantial changes in fire regime are indicated by the dotted vertical lines.

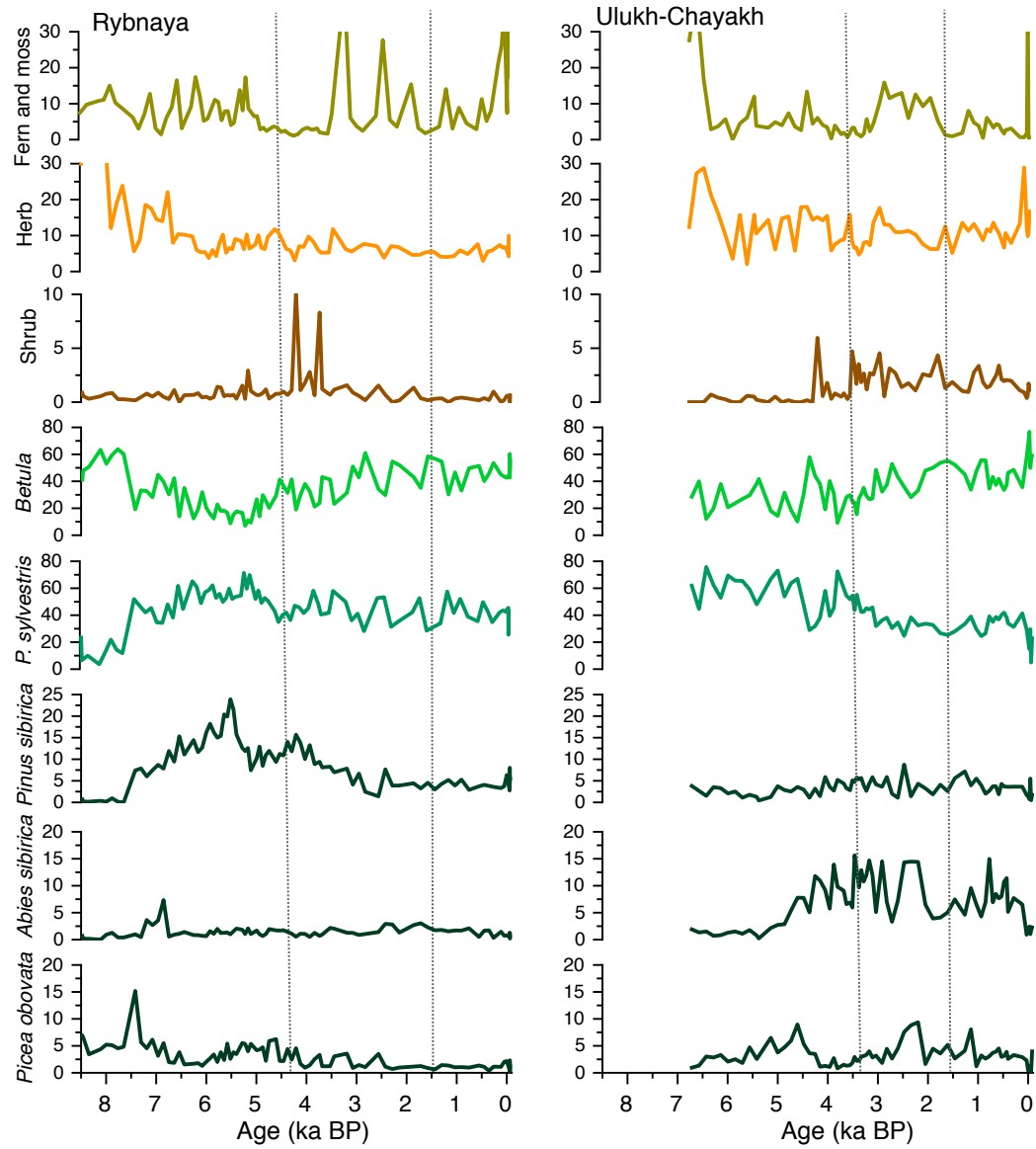

**Figure 4.** Temporal trends in the pollen percentages of the fire-related plant functional groups (invaders, resisters and avoiders) and the forest type determined from the ratio between pollen percentages of dark and light taiga tree taxa in the Rybnaya and Ulukh-Chayakh sequences. Hydrological conditions of the peatlands were derived from testate amoeba-based estimates of the water table depth using the European transfer function (blue) and the Asia transfer function (grey); where positive Z-score values represent greater-than-mean water table (deep water table) and negative Z-score values represent lower-than-mean

water level (shallow water table). The fire metrics include burned biomass (CHAR), fire frequency (number of fires /900 years) and charcoal peak magnitude (the higher the values, the greater the fire severity and/or closer to the site) derived from macro-charcoal particles >150 μm. Colours in the background denote periods with marked changes in fire regime and vegetation composition, where a change from blue to yellow indicates an intensification of the fire episodes.

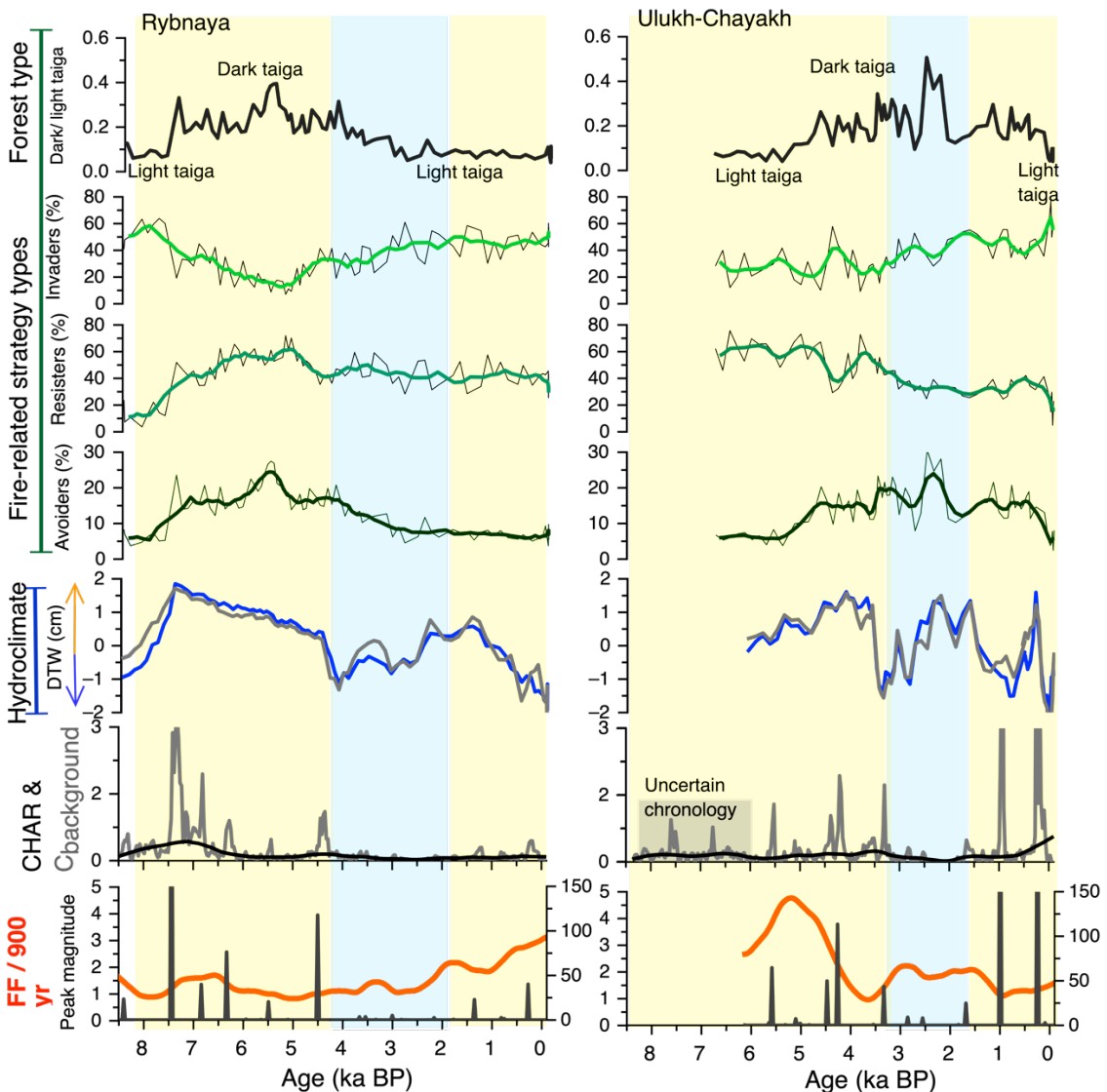

**Figures 5**. Charcoal based fire frequency at here locations in *Pinus-Betula* dominated boreal forests Russia: northern Ural region (Barhoumi et al., 2019), Lake Baikal (Barhoumi et al., 2021) and western Siberian Plain (Rudaya et al., 2020) (a). Composite record of biomass burning (n=4) based on Z-score charcoal influx where positive/negative Z-score values represent greater-than-mean/lower-than-mean charcoal influx over the base period (b). Composite record of fire frequency (n=4) (c). Composite record of peatland hydrology (n=4) from testate amoeba where positive/negative Z-score values represent

lower/higher-than mean water level (d). Annual temperature (anomalies) for 30-60 °N (Kaufman et al., 2020) (e). Composite

record (n=4) of the relative abundance of avoiders, resisters, and invaders determined from pollen percentages (f). Grey curves represent confidence intervals. Colours highlight the periods with higher biomass burning in the composite record.

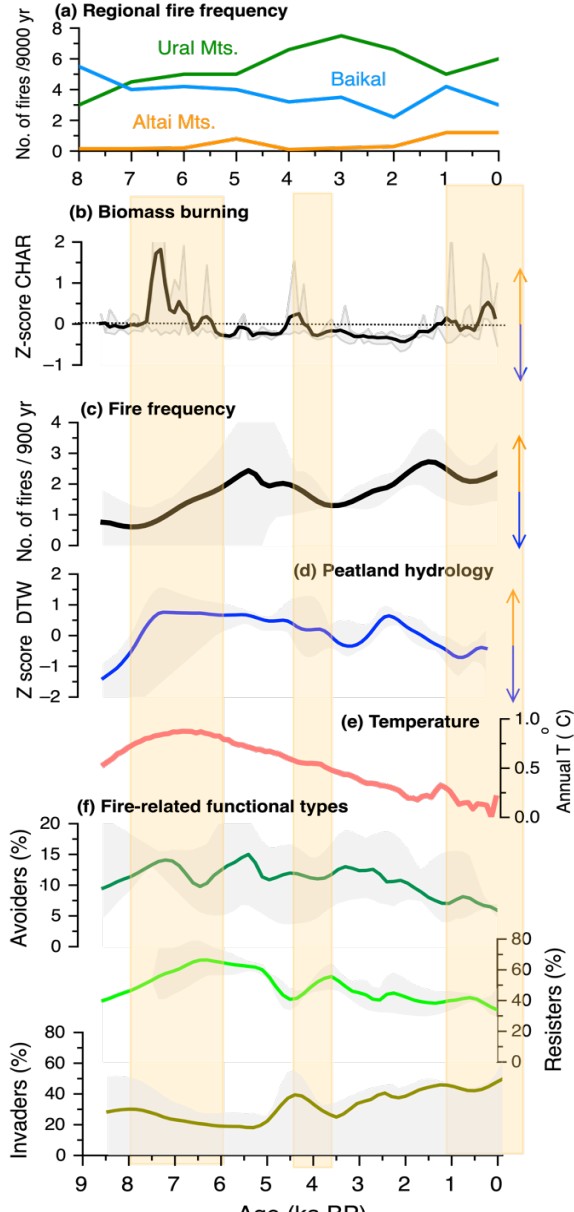

**Figure 6.** Correlation analysis between water table position, the main plant functional types, and the light-to dark taiga index (forests density) and micro and microcharcoal for Rybnaya (a) and Ulukh-Chayakh (b).

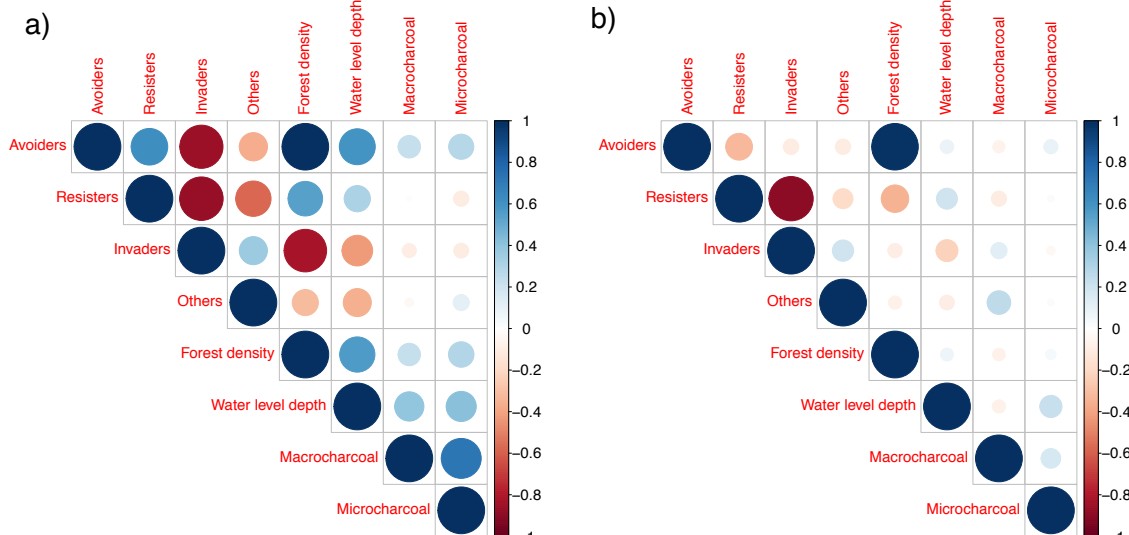

