# Peer review of "Holocene wildfire regimes in western Siberia: interaction between peatland moisture conditions and the composition of plant functional types"

_Climate of the Past, 2021_

## Author Response (AR1)

Frankfurt am Main, 16.01.2022

Dear Editor, Nathalie Combourieu Nebout

Please find enclosed a revised version of my manuscript entitled *'Holocene wildfire regimes in forested peatlands in western Siberia: interaction between peatland moisture conditions and the composition of plant functional''*
We have addressed and incorporated all comments made by the reviewers. Our detailed response is marked in colour.

We thank the reviewers for their valuable comments that helped to improve the current version of the paper and for encouraging words about our work: We hope that you will approve the revised content of the manuscript.

Kind regards
Angelica

**Anonymous Referee #1 (text in red)**
Referee comment on "Holocene wildfire regimes in forested peatlands in western Siberia: interaction between peatland moisture conditions and the composition of plant functional
types" by Angelica Feurdean et al., Clim. Past Discuss., https://doi.org/10.5194/cp-2021-125-RC1, 2021
*This is a review for the manuscript "Holocene wildfire regimes in forested peatlands in western Siberia: interaction between peatland moisture, vegetation composition and fire regime from two peatland sites in Western Siberia. The study provides new and valuable insights of the relationship between the different proxies and the significance of the findings for ongoing and future climate change in boreal peatland ecosystems. Overall, I find the paper well-structured and the multiproxy data clearly presented with sound interpretations of the paleorecords. However, I would encourage the authors to make some of the fonts in figures larger to be print-friendly. I am not an expert for amoeba and thus my review comments are focussed primarily on the paleofire and vegetation records, which align best with my own research background.*

R: We thank the reviewer for valuable comments that helped to improve the current version of the paper and for encouraging words about our work.

**General comments**
*Introduction: I think the authors mix fire (events) with fire regimes a few times and I suggest checking that the words are used correctly as fire regime is a defined term describing the general pattern in which fires naturally occur in a particular ecosystem over an extended period of time not an individual fire. See e.g., line 65 ff.*
R: Thank you for this observation. In the revised version, we have replaced the term fire regime whenever this was incorrectly used.

*Chronology: I would like to see more details on the chronology establishment in the main text, for example what material was dated and also why the authors included so many bulk dates instead of picking terrestrial macrofossils for dating purposes. The site description with local Betula growths implies that the record should contain enough datable terrestrial macrofossils for 14C purposes thus it's difficult to understand, why bulk material was dated instead.*
R: In the revised manuscript, we have included additional information on the plant material dated. However, accommodating the comments from 4 reviewers has led to a substantial increase in the length of the manuscript so we have added the information on the chronology in the SI. See File S1, l. 1-60.

*Charcoal morphology: The authors state that they grouped charcoal in woody and nonwoody at both sites based on morphology criteria. However, they also state that the charcoal was classified based on length ratio for one site to achieve an additional classification of graminoids vs leaves/wood. It is unclear why they chose two methods to achieve the ratio for one site while only one method for the other site. This is especially surprising since the two classifications largely overlap for interpretation (graminae vs. woody/leafs and non-woody vs woody).*
R: Our standard method to determine charcoal types was via charcoal morphological classification which can identify a series of fuel types (see Fig S3A). At a later date, we additionally tested the length to width ratio method (L:W), which is a more simple way to determine fuel type. The reason that the L:W ratio method was only tested at UC site, is because we did not have peat material left at Rybnaya. In the process of charcoal morphological identification, we often break charcoal particles, thus measuring the length and width on samples already used for charcoal morphological identification would have likely introduced errors. We, therefore, refrained from testing the L:W ratio method at Rybnaya.

*Numerical analyses: I am a bit surprised to see how local indicators (watertable, macroscopic charcoal) and regional indicators (pollen) were combined for correlation calculations. Usually, microscopic charcoal >10um is used for*

*regional reconstructions that would align better with the catchment of micron-size pollen. The authors should consider adding a justification why in this case such correlations across spatial scales are considered more useful than combining proxys with a similar catchment estimate? The difference in the geographic scale covered by the individual proxies seems also the most likely explanation why the authors couldn't find some of he statistical correlations they were expecting between watertable, fire and vegetation indicators*

R: We kept the former indicators but also included microcharcoal fraction in statistical analysis. We also discuss the differences resulting from the various spatial scales of our proxies.  See l. 170-171; l. 284-290; 428-435.

*Comparison between recent charcoal and satellite images: In my opinion the first paragraphs are presenting results rather than discussion of the results and should be moved to the results section. For the discussion of the relationship between charcoal based fire reconstructions and satellite-derived fire detection, I would like to see some references to other studies such as e.g. Adolf et al. (2018) or Daniau et al. (2017)*

R: We have moved a part of this chapter at the Results under a new heading reading: ''3.2. Fire-type identification from charcoal, satellite images, and forest statistics'' l. 220-234, and kept the one paragraph in the Discussions under:  4.1 Changes in fuel, fire type and fire frequency over the Holocene, l. 295-304. References suggested were also included.

*Betula pollen: Please clarify if you determined tree type Birch and shrub type birch (Betula nana) pollen (e.g. see Birks 1968). If Betula could not be identified, it may contain Betula nana as well, not only Betula trees, which should be considered for discussing the results.*

R: Added: '' The two tree birch species (*B. pubescens* and *B. pendula*) and the dwarf shrub (*B. nana*) could not be differentiated by pollen analysis and were considered as *Betula* spp. However, at present day, *B. nana* tends to be found only sparsely in this region, l. 174-177.

Specific comments

*Line 64-65 Reference needed: When these aerosols persist in the atmosphere, it leads to a medium-term increase in albedo and ultimately to regional cooling.*

R: This sentence links to the citation above: Rogers et al., 2015.

*Line 65ff: The sentence needs rewording, it currently reads : fire regimes ... are surface fires, see general comment above*

R: Thank you, we have reworded this sentence: '' Fire types in the Siberian boreal forests are often litter-fuelled surface fires …''.l. 68.

*Line 97: add spore analysis to pollen analysis*

R: Revised to: ''In this study, we used multi-proxy analyses (pollen, non-pollen palynomorphs, microcharcoal, macrocharcoal morphologies, testate amoeba, geochemistry) on two…''. l. 101-102.

*Line 99: I think it would be better to add "plant/vegetation, ...)" to "community" to clarify what community the authors refer to*

R: Revised to: '' We have determined how peatland moisture has interacted with tree community composition with regard to plant functional types and fuel flammability in driving the frequency and type of fires'' l. 104-105.

*Line 103: the term "charcoal site selection" seems misleading as the authors conduct a multiproxy paleoecological study not just charcoal analysis*

R: Agree and rephrased to: ''Geographical location and site selection''. l.108

*Line 112: I believe the coring was conducted near the river not on the river*

R: Thank you, revised to ''We cored two sites, Rybnaya Mire (N57,27566800°, E84,48758900°) located near the Rybnaya river…'' l. 107.

*Line 114: Local peatland? Vegetation?*

R: Revised to: The local peatland vegetation at both coring sites includes mesotrophic open sedge-*Sphagnum* communities with small birch and pine trees at Rybnaya Mire and standing dead *Pinus* tree trunks at Ulukh-Chayakh Mire…'' l. 119-121.

*Line 115: What species are the dead tree trunks?*

R: *Pinus*, but we do not know which species. See our response above.

*Line 124: add unit (cal yr BP) to the surface age -69 and -67*

R: Added: '' In the age-depth models, we assigned a surface age of -67 cal yr BP (coring year 2017 for Rybnaya) and -69 cal yr BP.'' l. 129.

*Line 129: fire history is not inferred from the peat sample but from the charcoal information in these samples. This needs to be rephrased.*

R: Rephrased to: ''We inferred changes in local-scale fire regime based on charcoal particles identified in peat samples of 2 cm$^3$ extracted at 1 cm contiguous intervals''. l. 134-135.

*Line 137: Reference for charcoal catchment. Consider for example Adolf et al., 2018*

R: Added. L. 143.

*Line 230ff: provide absolute values of change of understory vegetation values rather than "more", "predominantly"*

R: Revised to: 'Understory vegetation primarily reflects local peatland development. It was predominantly composed of herbs/sedges (Cyperaceae; up to 20%) and ferns (Polypodiaceae, up to 20%) and *Equisetum* (up to 25%) at both sites until ca. 4.5 ka. An increase in the abundance of shrubs (Ericaceae; up to 10%) and moss (*Sphagnum;* highly fluctuating between 10 and 100%) at Rybnaya, and Ericaceae (up to 5%) and moss (*Sphagnum;* up to 100%) at Ulukh-Chayakh was noted over the last 4.5 ka (Fig. 3; File S4a,b). '' l. 264-270.

*Line 240f: mention water table values as well for the second part of the paragraph rather than relative descriptions.*

R: Added. 'At Ulukh-Chayakh, the water level was high (around 15 cm below surface) between 3.5 and 2.5 ka, 2 and 0.5 ka, and towards the present day. There, times of low water table (34-20 cm below surface) occurred between 6 and 3.5 ka, 2.5 and 2 ka, and around 0.5 ka''. l. 277-279.

*Line 244ff: the lack of correlation could be due to the different proxies reflecting local and regional spatial scales.*

R: We have run an additional analysis on microcharcoal fractions and discussed the influence of spatial scale on the statistical output. See the Chapters 3.6. Numerical analysis, l. 284-290 and 4.3 Feedbacks between peatland moisture, community PFT composition and fire regime, l. 428-435.

*Line 265f: charcoal occurrence should be presented as concentrations of particles per volume or better as charcoal influx per year and a given area rather than "pieces".*

R: Added, please see chapter 3.1. l. 228-234

*Line 289: add "for both sites"?*

R: Added.

*Line 320: determination of the detrital element Ti should be mentioned in the methods section*

R: The geochemical analysis section is now presented as an individual chapter at the Methods. See 2.6 Elemental geochemistry. l.295-300.

*Line 401: Scenario should be replaced by period if considering how it is referred to in the following sentences.*

R: Revised to: ''At a Holocene timescale, we found two periods with contrasting moisture-vegetation-fire interactions''.l. 478-479.

*Line 411ff: references needed for future climate scenarios in Siberia*

R: IPCC, 2021, l. 490

*Appendix A1: font too small to read when printed in A4;*

*Figure 6: labels for correlation: increase font size.*

R: Done

*Figure 1: fonts are very small in the maps, indicate latitudes and longitudes as well as direction of North for map b-d;*

R: Done see revised Fig. 1

*All figures showing Holocene records: x-axis, I believe you show kilo-years here not years, adjust unit accordingly*

R: Fonts enlarged and cal yr BP changed to ka BP.

Technical comments

*Line 64: this leads =they lead*

R: Corrected, l. 66

*Line 97: Multiproxy analysis should be used in plural form here*

R: Replaced with: Multiproxy analyses. l.101

*Line 99: "with regard to" instead of "in"*

*R:* Replaced with ''…with regard to…''. l. 104.

*Line 122: add supplementary to file S1*

*Line 171f: using vs used, somehow the sentence structure is wrong*

R: Revised to: ''This was done by averaging the Z-scores and smoothing them to 300 years with a locally weighted regression and a 95% confidence envelope on the same sites used for composite biomass burning reconstructions in the R palaeofire package''. l. 182-184.

*Line 354: change primally to primarily*

R: Corrected.

*Line 378: "from" or similar word missing*

R: This sentence was removed altogether.

Comment on cp-2021-125

**Anonymous Referee #2 (blue)**

*This manuscript entitled "Holocene wildfire regimes in forested peatlands in western Siberia: interaction between peatland moisture conditions and the composition of plant functional types" and submitted to Climate of the Past by Feurdean et al., seems to be an original study, providing new elements of thinking on the functioning of borelean forests, in relation to the dynamics of fires, the climate and vegetation on peaty terrain.*

R: We thank the reviewer for valuable comments that helped to improve the current version of the paper and for encouraging words about this work.

*Here are my general comments:*
*- Regarding the layout of the figures, I think there are many: it might be wise to assemble them together, like figure 4 and 5 for example and to grade the same logic: either to everywhere do 1 panel per site, ie superimpose the 2 sites, but for each figure.*
R: Thank you. We have adjusted figure 3 after figures 4 and 5.

*- The discussion seems to me to be well constructed, however I find it unfortunate not to have carried out a specific discussion figure, in particular regarding the dynamics of fires, since Feuredean et al., cite several studies on the scale of Siberia but also of Russia, it would have been beneficial to put their results directly in comparison with those cited, within a summary figure.*
R: We have included a new panel in Fig. 5 (a) comparing the FRI in Pinus-Betula dominated taiga in Siberia to our FRI reconstruction. Please see the revised Fig. 5.

*- I find it more prudent to nuance the conclusion about the water table, given that on the 2 sites studied, only 1 showed a significant result indicating that the fire regime was greater from the 20 cm threshold. And I will add to the conclusion that further studies from different sites would be necessary to regionally confirm this.*
R. Agreed and revised as: Our results indicate an increased sensitivity of mesotrophic peatlands to burning at water levels below 20 cm, however, more regional studies are necessary to determine thresholds in peatland moisture to fire. l. 594-495.
*Here are more specific comments:*
*L38: "Pinus sylvestris-Betula"*
R: Replaced *Pinus- Betula* with *Pinus sylvestris-Betula*. l. 38
*L60: references are missing*
R: Wildfire is the most common type of disturbance in boreal forests (Kharuk et al., 2021 and refs therein). l.61.
*L76: You can not write "wildfire regime" and then include human activities within. Rather use just "fire  regime".*
R: Thank you, rephrased to: ''A fire regime emerges from the combination of ignition sources, climatic conditions, fuel properties and human activities...''. l. 78.
*L176-109: the meaning of the sentence was not very clear, I would suggest: "In this region, the forest is made up of both light taiga (...) but also dark taiga (...), in greater proportions.*
R: Rephrased to: 'Regionally forest is made up of both light (*Pinus sylvestris, Betula pubescens, B. pendula, Populus tremula*) and dark (*Pinus sibirica, Picea obovata,* and *Abies sibirica*) taiga, the latter in greater proportions towards the east (Laschinsky and Koroliuk, 2014). l. 113-115.
*L659: I would change "grey rectangle" by "grey time windows".*
R: Changed. l.814.

**Sergey Loyko (lilac)**
**General comments**
*Angelica Feurdean's article is a complete, multi-proxy study that is extremely important and interesting for understanding the history of the boreal forests of southeastern Western Siberia. The article is the result of the work of a large team over several years. For the most part, the article is already ready for publication; it is also of interest to Russian scientists engaged in related research in the south of Western Siberia. However, before publishing the article, I would recommend making some changes, both in the logic of the interpretation of the results, and making minor technical changes. https://disk.yandex.ru/d/8XbeP1vRk_y4lQ*

We thank the reviewer for valuable comments that helped to improve the current version of the paper and for encouraging words about this work.

*Populus tremula is a tree species that does not leave traces in the spore-pollen spectrum. However, this species is abundant in the taiga of the Tomsk Region, which is clearly visible on the map: https://disk.yandex.ru/d/8XbeP1vRk_y4lQ (osina, green). If the community of dark coniferous species is destroyed (as a result of fire or insect infestation), they are replaced by birch, aspen or pine. Aspen can form large monodominant forests. Moreover, the existence of such communities can reach several hundred years if the rudiments of dark conifers do not come from outside. Moreover, if there are no dark conifers, then gradually aspen forests can replace birch and pine forests (Pinus sylvestris). That is, aspen forests can have a significant impact on the proportion of birch and pine pollen. This is worth mentioning in the discussion.*

R: Thank you, we have revised the text as follows in the Methods: ''Due to its poor pollen preservation, *Populus,* a fire endurer, is mostly absent from our pollen records. The necessity to combine the *Betula* pollen signal and the paucity of pollen of *Populus* may have resulted in the endurer PFT group failing to be represented in our records'' l. 175-177'' and Discussion: ''The post-disturbance (fire, insect outbreak) forest regeneration pathway in Siberia also involves *Populus tremula.* Eurasian aspen forest communities can reach hundreds of years in age and change forest composition to *Populus* dominance where regeneration of dark taiga conifers is absent (Tautenhahn et al., 2016; Kharuk et al., 2021). Although we have not encountered *Populus* in our pollen record due to its poor pollen preservation, it cannot be discounted that *Populus* may have been a part of post fire forest succession pathways in the past''. l. 464-469''.

*In determining the fire regime, an important role is played by how close the swamp is located to the mineral bank. Fire usually enters the swamp from the mineral shore. Also, the presence of a cover of sphagnum mosses affects the possibility of fire spread. The latter severely limit the possibility of bog burnout. Based on Figures S3b and Figure 4, it can be seen that the sphagnum mosses appeared on Rybnaya about 3600 years ago, after which the peat fires disappeared. The sampling point itself is quite distant from the mineral shore of the bog, a couple of kilometers (excluding isolated mineral forest islands). This further reduces the likelihood of a peat fire on Rybnaya. The Ulukh-Chayakh point is located just 160 meters from the mineral shore, sphagnums began to play a role only in the last centuries due to the large fires of the Russian time. As a result, peat fires occurred on Ulukh-Chayakh until recently (less than 500 years).* These differences between the sites under consideration should be reflected in the discussion. *A good example of how the number of fires decreases with distance from a mineral shore, is the data from the article: Turunen J., Tahvanainen T., Tolonen K., Pitkänen A. Carbon accumulation in West Siberian mires, Russia // Global Biogeochemical Cycles. V. 15 (2). P. 285– 296. DOI: 10.1029/2000GB001312 Note that in the last article, the sources of fires are anthropogenic.*
*Obviously, a fire in a swamp is not only a factor in changing the vegetation cover, but also strongly transforms hydrological parameters. Obviously, bog fires could also affect the level of bog water, in case of damage to the stand, or the introduction of other plant species. This point seems to me important and, at least, it makes sense to mention the possibility of such an effect.*
R: Thank you, we have revised this section of the manuscript according to your suggestion:'' Mire type development and changes in peat plant communities could have also influenced peatland hydrology (Galka et al., 2016; Kurina and Li, 2018; Blyakharchuk and Kurina, 2021). At Rybnaya, the minerogenic stage of the mire occurred from 8 to 4.5 ka and the water level was more stable. With the meso-oligotrophisation of the mire, indicated by the appearance of Ericaceae at 4.5 ka, and the dominance of *Sphagnum* from 3.6 ka to the present, the water level and the amplitude of its fluctuation rose and the occurrence of peat fires declined (Figs. 3, 4; File S4a,b). At the Ulukh-Chayakh site, Ericaceae developed at 4.5 ka and remained dominant up to the present, whereas *Sphagnum* established around 2 ka and became dominant only in the past decades (Fig. S4). Further, mire margins with thinner peat depth burn more frequently than thicker peat (Turunen et al., 2001). Thus the location of the Ulukh-Chayakh sampling point close to the mineral margin of the mire (ca. 160 m), as opposed to a few kilometres at Rybnaya, could have sustained a high fire occurrence at this site until recently. Lastly, fire itself can influence peatland hydrology, with field measurements indicating an age-dependent water table lowering after burning (Holden et al., 2015). It is beyond this study to discuss all the details of the drivers behind peatland hydrology, however, given our main focus is the link between state changes in peat hydrology and fire occurrence and severity. Please see l. 352-363.

***Today it is known that the Neolithic Settlement of Kayukova-2 is located there. Note that the peaks of fires on Rybnaya correlate with the heyday of the Neolithic (7-5.5 years ago) and Eneolithic (5-4 thousand years ago) cultures. At UC, there are correlations with cultures from the Bronze Age and the Middle Ages. However, there are no publications with reliable radiocarbon dates, so this is still only a hypothesis. On Rybnaya, most likely, there has been no peat fire for the last 4.5 thousand years*. *On Ulukh-Chayakh, they continued until recently.***
R: Many thanks for providing these references, we have included archaeological finds and the potential connection of human impact with fire and vegetation in the revised manuscript: ''A GIS survey of archaeological finds in the Tomsk region reveals a high density of sites, but no distinctions between cultural phases of these sites exist (Zolnikov et al., 2020). In parallel with modern settlement distribution, most of these ancient sites were situated near rivers favouring the floodplain terraces of river valleys (Zolnikov et al., 2020). Another study in the Lower Tom basin attests to the presence of tools that have the characteristics of sites commonly associated with the Neolithic and Bronze Age close to Rybnaya (Idimishev et al., 2018).'' See l. 337-411.

*The disadvantage of this work is that the results of work on this topic, which were carried out by scientists from Russia and Tomsk, are not reflected in any way. Many are in Russian, but Russian co-authors could help translate these articles. I recommend correlating the results obtained with the following articles:*
*(Blyakharchuk et al., 2003): doi: 10.1191/0959683603hl658rp (Borisova et al., 2011): doi: 10.1016/j.quaint.2011.01.015*
*(Willis et al., 2015): doi:10.1177/0959683615585833*
*(Kurina, Blyakharchuk, 2020): doi:10.1088/1755-1315/611/1/012025*

(Willis et al., 2015): doi:10.1177/0959683615585833
(Krivonogov et al., 2012): doi:10.1016/j.palaeo.2012.02.030
(Blyakharchuk et al., 2019): doi:10.17223/19988591/45/9
(Blyakharchuk et al., 2018): doi:10.17223/19988591/42/12

R: Thank you for providing us additional literature from the study region. We have added most of these references into the revised paper.

Specific comments
Abstract:
*Line 40: It is better to use deciduous forests instead of "broadleaf forest". In Siberia, there are practically no broad-leaved species, except for small areas of linden, as well as plantings in cities.*
R: Broadleaf replaced with deciduous forests.

*Lines 44-48: It is necessary to reformulate these results. So we are talking about the intervals between 7.5-4.5 ka BP and later.*
R: We have altered this slightly to: '' On a Holocene scale, we found two periods with contrasting moisture-vegetation-fire interactions: (1) severe fires were recorded between 7.5 and 4.5 ka BP with lower water level and an increased proportion of dark taiga and fire avoiders (*Pinus sibirica* at Rybnaya and *Abies sibirica* at Ulukh-Chayakh) with a predominantly light taiga and fire-resister community characterized by *Pinus sylvestris*; (2) Severe fires occurred over the last 1.5 ka and were associated with fluctuating water tables, a declining abundance of fire avoiders, and an expansion of fire invaders (*Betula*).''. l. 44-48.

*Line 95: Articles "Rudaya et al., 2020" on the steppes. You can also add the following article to this paragraph: Lamentowicz, M., Słowiński, M., Marcisz, K., Zielińska, M., Kaliszan, K., Lapshina, E., Gilbert, D., Buttler, A., Fiałkiewicz-Kozieł, B., Jassey, V.E.J., Laggoun-Defarge, F., Kołaczek, P. Hydrological dynamics and fire history of the last 1300years in western Siberia reconstructed from a high-resolution, ombrotrophic peat archive (2015) Quaternary Research (United States), 84 (3), pp. 312-325. DOI: 10.1016/j.yqres.2015.09.002.*
R: Location corrected. The study by Lamentowicz et al., 2015 did not look at the fire regime in a statistical sense as the other papers cited here did. However, we have mentioned this paper in the Discussion when looking at fire-peatland moisture-species interaction. See l. 430.

*Line 105: What is the Subarctic Climate Impact? This is the first time I hear about it. This can be said about any boreal region, so it is better to remove it. You can also replace "continental climate" with "continental boreal climate"*
R: Replace to: ''The region has a continental boreal climate''. l.110.

*Line 108: Light and dark taiga are copyright terms, you need to indicate it here. Populus tremula is also included in the light taiga.*
R: Rephrased to: 'Regionally forest is made up of both light (*Pinus sylvestris, Betula pubescens, B. pendula, Populus tremula*) and dark (*Pinus sibirica, Picea obovata,* and *Abies sibirica*) taiga, the latter in greater proportions towards the east (Laschinsky and Koroliuk, 2014). l. 113-115.

*Line 107-109: This information is not available in the cited sources (Berezin et al., 2014; Rybina et al., 2014). Although there is practically nothing to find in English, there is a lot of good research in Russian. The following article by N.N. Lashchinsky with a large English abstract: https://elibrary.ru/item.asp?id=24116649*
R: Please see above.

*Line 115: How did you know that birches are young? Was their age determined? Maybe they are just short, dwarf? Wasn't Pinus sylvestris there?*
R: The birch and pine trees at Rybnaya were small, but we agree to the fact that being small does not mean being young, but perhaps dwarfed. Rephrased to: ''The local peatland vegetation at both coring sites includes mesotrophic open sedge-*Sphagnum* communities with small birch and pine trees at Rybnaya Mire and standing dead *Pinus* tree trunks at Ulukh-Chayakh Mire.''l. 119-121.

*Line 219 and further: How was forest density determined? The fact of the prevalence of "light taiga" in the vegetation cover of plant communities does not mean that the forest was of low density (low projective cover?). Birch or aspen forests can have a fairly high crown density, forming a high density forest. It is necessary to clarify this term and its calculation.*
R: In fact we determined the ratio of dark-to-light taiga, an approximation of forest density, however we agree that light taiga birch forests can have high crown density, as opposed to Pinus sylvestris. Revised to: ''We also separately grouped the tree pollen into taxa associated with dark taiga (*Pinus sibirica, Abies sibirica, Picea obovata*) and light taiga (*Pinus*

*sylvestris, Betula* spp*., Larix* spp.) and created an index for dark-to-light taiga composition, an approximation of forest density, by calculating the ratio between the relative abundance of dark and light taiga tree taxa (Higuera et al., 2014).''.
l. 179-183.

*Line 255: During riding fires in Siberia, standing tree trunks, as a rule, only burn. Usually, after a fire, standing burnt trunks remain, which then can fall for several decades. Crowns, bark, litter and trunks that have previously fallen to the ground burn and burn completely.*
R: Thank you, however such differentiation would be difficult based on the fossil records.

*Lines 331-335: After the appearance of sphagnum on Rybnaya, the mire became more watered. That is why it could burn less, since after the appearance of sphagnum, based on figure S3b, there are no more large charcoals here. On Ulikh-Chayakh, this is not observed and fires continued to enter this mire from the mineral shore.*
R: We agree, please see our response above and under l. 264-270 and 352-363.

*Line 346: Is peatland moisture primary? Maybe the humidity of the climate is primary? Do you interpret peat moisture in terms of climate moisture? Or do you think that this is only a consequence of changes in hydrology associated with the self-dvelopment of the peatland? In my opinion, both are captured in the water level curve.*
R: We primarily connect peatland hydrology with climatic conditions i.e., temperature and/or precipitation. We have briefly extended our discussion on competing drivers of moisture changes (l. 352 -383), as well comparison with regional reconstructions of changes in moisture conditions (l. 354 -394).

*Line 350: I am not aware of any cases of B. pendula growing on peat soils in the Tomsk region.*
R: This is a misunderstanding, we have not state that it does occur on peat soils but on sandy soils.

*Lines 375-376: Check the correctness of the quotation. I saw that in an article by Kharuk et al. (2021) says the opposite, for example: "It is expected that more frequent and severe fires will promote substitution of the DNC within their southern range by broadleaf (birch and aspen) and drought-resistant larch and Scots pine species".*
R: Yes, the text above highlights the most common pattern in Siberia, proposed by Kharuk et al., 2021, but this paper also states that ''Fire also allows dark taiga species to compete with *P. sylvestris* and become established beyond poor soils and boggy areas''. We therefore retained this sentence.

*Line 381: Calluna vulgaris does not grow in the studied region. Heatherland is also not typical for the south of the Tomsk region. In the southern taiga, forests are herbal. Species of the Ericaceae family have large coverage on (1) sandy mineral soils; (2) in the peatlands. In the forests adjacent to the Rybnaya site, sandy soils, on Ulukh-Chayakh loamy. The contribution of ericoid pollen from forests on sandy soils near Rybnaya should be small, due to the remoteness of the bog from the sand dunes. For the Ulukh-Chayakh site, ericoid shrubs are typical for bogs, since forbs prevail on the nearby mineral island. In this regard, we can say that the appearance of ericoid pollen is mainly associated with the evolution of bogs. On Rybnaya, ericoids appeared after 4500 ka, and oligotrophization began. Then the content of ericoid pollen decreased, as the invasion of sphagnum mosses took place, which reduced the coverage of ericoids.*
R: Yes, this was also our interpretation, we have now extended this interpretation. Please see l. 264-270 and l. 352-363.

*FigS5 and Lines 321-322: There are two oddities in the distribution of Ti over the Rybnaya peat deposit. The first is the increased concentration of titanium in a layer about 50 cm thick from the base of the peat deposit. The second oddity is the lower Ti concentration at the bottom of the peat deposit, compared to the overlying 50 cm layer. Also, in Table S1 and Figure S1, date inversion is noticeable. In my opinion, this is a consequence of either the windblown tree, or the action of the stream, now buried under the peat. This can be mentioned in the interpretation. Further, we note that the Ti peak in the UC sequence between 4 and 3 ka approximately corresponds to an inversion of the C14 age at a depth of 185 cm (Rosaceae seeds); perhaps this is not an accidental coincidence. It is likely that this is the influence of a powerful flood in Chulym, which flooded the first terrace above the floodplain, on which mire UC is located. There are indications of powerful floods in the period 4-3 ka in Russianlanguage literature (for example, Leshchinskiy, S. V., et al. "The first terrace above the Ob'floodplain near Kolpashevo: the age and formation conditions." Russian Geology and Geophysics 52.6 (2011): 641-649).*
R: Thank you for the information and references on flood events at UC, we have taken this into account. Please see our extended interpretation in File S1.

Technical corrections
*Figure 1: Site Ulukh-Chayakh on the map is marked between deciduous forest and mire. But in fact, on the left, there is a mineral shore, on which there is an abandoned arable land. It is necessary to correct the contours on the map, since this contact, in my opinion, is important for the interpretation of the data.*
R: Done, please see revised Fig.1

*Lines 19-21: Fix affiliations. Affiliation errors number 7 and 9.*
R: Corrected.
*Line 37: "... near Tomsk Oblast, Russia" replace with "... from Tomsk Oblast, Russia".*
R: Corrected to: ''To address this knowledge gap, we reconstructed the Holocene fire regime, vegetation composition and peatland hydrology at two sites in Western Siberia in the Tomsk Oblast, Russia''. l. 37.
*Linu 112: In degrees, correct the commas for periods.*
R: Corrected, l. 177-118.
*Line 114: About 30-40 km to the border.*
R: Corrected to: ''…located on a terrace of the Chulym river, near the village of Teguldet, about 30-40 km from the border of the Krasnoyarsk Krai''. l.118-119.
(*Line 166: Missing reference to Gill, 1981.*
R: Added in ref list.

*Community commetns 4: Irina Kurina (orange brown): The manuscript "Holocene wildfire regimes in forested peatlands in western Siberia: interaction between peatland moisture conditions and the composition of plant functional types" by Angelica Feurdean et coauthors is of great interest and raises a very relevant topic for scientific research. This research is based on a variety of evidence, it is used a multi-proxy approach to the detailed palaeoenvironmental reconstruction. Age-depth models for the studied peat cores as a necessary basis for work looks very strong. I think also that the study area related to the southern part of Western Siberia, on the one hand, is poorly studied and, on the other hand, this is the one of the most suitable places in Eurasia for such research because of the widespread occurrence of peatlands. Although the design of the manuscript (so called paperwork) leaves much to be desired. And the interpretation of the data obtained raises many questions. Therefore below I provide my comments (questions) to the manuscript.*

R: Thank you for the valuable comments that helped to improve the current version of the paper.

*Line 58 – I think it is unclear and you should explain how climate-disturbance-fire feedbacks do affect overall future resilience of forested peatlands to climate change.*
R: This is stated in the prevision sentence, which reads: ''However, dry conditions, particularly a water table decline below the threshold of 20 cm, will probably exacerbate the frequency and severity of wildfire, disrupt conifers' successional pathway, and accelerate shifts towards more fire-adapted broadleaf tree cover''. We do not consider that this needs further explanation in the Abstract. Details can be found in the Discussion.

*Line 175 – I think this is incorrect heading for paragraph. You cannot apply directly reconstructed water table depth based on testate amoeba data to climate reconstruction without any explanations. In fact, testate amoebae are protists, which inhabit the upper layer of cover in peatlands. They indicate local conditions of surface wetness in a peatland. And the level of this locality is very small, which means that there are only a few centimeters around. Based on testate amoeba data from only one place (microsite) in a peatland we cannot judge surely even about the degree of waterlogging for this peatland of itself, in general. Moreover we cannot say anything about climate (hydroclimate). Before using the reconstructed DWT for climate reconstruction, you should show the clear connection between the two.*

R: We have provided basic information on the methodology of TA identification and depth to water reconstruction. We agree that the link between the water table and climate needs to be further demonstrated. However, it is beyond this study to extend on the drivers of peatland moisture changes but rather the link between state changes in peat hydrology and fire occurrence and severity. The evidence provided by our results is sufficient to suggest a link between peatland hydrology and fire activity, specifically dry peatland conditions are likely to burn often and /or severely. We have extended the explanation on testate amoeba hydrological reconstruction slightly as well as the comparison with reconstructions of changes in moisture conditions showing synchroneity in the hydrological conditions in the region. Please see our revised lines 346-390; 428-435. The full record of testate amoeba (TA) assemblages as well as the hydrological reconstruction and interpretation at the four sites will be published extensively in a future publication (Diaconu et al., in prep).

*Lines 53-58 – you express your suggestion about possible future changes in forested peatlands in according to climate changes in the Abstract section. Next, Lines 411-417 – you mention about this suggestion in Conclusion section, but I have not found anything about this idea along the Result and Discussion sections of your manuscript. It is incorrect, when new thoughts appear in the Conclusion section, if they are not discussed before in the Discussion section or they are not mentioned in the Result section of the manuscript. I think it would be better and logical to expect moving this suggestion from the Conclusion to the Discussion section and adding the references to the published scenarios of climate changes in the future, that you mention in Line 54 and Line 412 of your manuscript.*

R: These are predictions for potential future forest-fire interactions as a response to future climate changes, particularly in peatland hydrology based on what we have learned from our palaeoecological data. We do not consider that this information comes out of the blue, but rather it follows logical findings from our long-term records. We, therefore, decided to keep these here.*Line 35 – you write in the Abstract "... despite their huge extent in boreal regions". This is one of the key messages that emphasizes the importance and relevance of your research. Please add some phrases about this in the Introduction section and show the facts (figures), what is the area covered by forested peatlands in Eurasia, in West Siberia and/or in the southern border of taiga zone in western Siberia. You can take this information from different published papers and books (as examples, 1) Vompersky et al. 2011 in Contemporary Problems of Ecology; 2) Kremenetski et al. 2003 in QSR; 3) Liss et al. 2001 monograph Wetland systems of Western Siberia and their conservation value – in Russian; 4) Alekseeva et al. 2015 in Bulletin of Tomsk Polytechnic University – in Russian).*

R: Thank you, we have added a sentence acknowledging this. It reads: '''Despite that Siberia contains a large extent of forested peatlands, particularly its western part (Vompersky et al., 1994; Liss et al. 2001; Kirpotin et al., 2021) no studies have explicitly explored the interactions between peatland moisture, vegetation composition, and fire regime in this region''. l. 98-100.

*Sorry, it is difficult for me to understand the key idea of your research. So I read the introduction of your manuscript. Lines 60-75 – you write about wildfires in forests. Next, lines 76-89 – you write about wildfires and peatlands. Next, line 90 – you write about fires in boreal ecosystems. Next, lines 91-94 – you write about forest ecosystems. Next, lines 95-96 – you write about peatland. Then, lines 97-101 – you express the aim of your research, but it is unclear if you are aimed to study forest or forested peatlands. You have said nothing about this. Although, in fact, forest and forested peatland are not the same. They differ. It is wrong to consider forests and forested peatlands as one and the same. Please explain what exactly you are studying – forests or forested peatlands. Along the manuscript the terms of forest and forested peatland are not separated. Especially in the title and in Conclusion section you declare the research of forested peatlands, although in the Result and Discussion sections you said firstly about forests, secondly about forested peatlands and thirdly about summary of forest and forested peatlands taken together (as one) without separation. As the result, the confusion between these three different things leads to the misunderstanding of the research and the interpretation of the data obtained.*

R: We started the introduction by acknowledging the role of fire in boreal forests generally (l.60-75). We moved to recognise that a lot of boreal forests grow on peatlands and that the relationship between such forests and fire is less known (l. 76-89). We then introduced the usefulness of palaeoecological research to capture past changes in fire regimes in forest ecosystems (l. 90-91). It is not relevant at this point whether a forest occurs or not on peatland, but understanding the dynamics of a forest ecosystem, which contains species that live decades to centuries, needs long-term records. Finally, the primary aim of our study is to look at forested peatlands. We believe that the introduction follows a logical path, however, in revising the paper we used the terms forests and forested peatlands more accurate.

*Line 170 – you write "To determine the regional changes in forest composition, we created composite records of PFTs". Could you explain, how do you determine the regional changes in forested peatland composition? How do you separate the composition of forests and forested peatlands? Based on my individual experience, I cannot imagine that tree composition is the same in a forested peatland and in forests, which surround it. In most cases they are different. As confirmation, let us look at the description of modern conditions in the studied peatlands (Lines 114-117): "The local vegetation at both coring sites includes mesotrophic open sedge-Sphagnum communities with young birch trees at Rybnaya Mire and standing dead tree trunks at Ulukh-Chayakh Mire". Then if we compare this with the upper samples from pollen diagrams related to the studied mires (Figs. S4a and S4b), we can see the other picture. Both the pollen diagrams show the great abundance of arboreal pollen, belonging mainly to Betula and Pinus sylvestris taxa. We can conclude, that, based on pollen data, there are forests, consisting of birch and Scots pine, but the studied mires are open, however you mention some birch trees on mire surface. It means that the composition of trees differs in forests and in forested peatlands. And pollen diagrams reflect the summary composition of forests around and of the tree cover in forested peatlands. Someone cannot separate in pollen diagram, where is the tree composition of forests and where is the tree composition of forested peatlands.*

R: We believe there is a misunderstanding of this sentence. It refers to results using the combined pollen records from the four peatlands (see their location in Fig 1) to obtain a regional picture of forest composition and dynamics. Each peatland we cored is forested and composed of tree species mentioned at 2.1 (please note a slight modification to accommodate Loyko's suggestion). The vegetation composition at each coring point was slightly open to make our coring easier, but otherwise, these peatlands are forested. Please see l. 428-.435.

*Line 320 – you used Ti concentration as "possible indicator of water influx". Could you provide any reference to the researches that confirm this idea? I am surprised to see such interpretation of the Titanium peaks. As far as I can consider from different papers (as example, Kempter and Frenzel 2008 in Science of the Total Environment; Margalef et al. 2014 in Palaeogeography, Palaeoclimatology, Palaeoecology; Hutchinson et al. 2016 in Regional Environmental Change – you cited the last reference) Ti is mainly precipitated from atmosphere. Its increasing peaks in a peat core (or lake sediments) can be caused by wind or soil erosion, by enchanced precipitation, or by increased production of the ecosystem. There are many reasons for positive peaks of Ti, but I never heard about river flood as the reason. If we look in your manuscript in Line 320 you write "The detrital element Ti, a possible indicator of water influx, was high in the bottom profiles that were rich in minerogenic material.". Then, in Lines 326-327 you write "Proxy records from Siberia attest to warmer and drier-than-present climate conditions between 9 to 6 ka ... (Groisman et al. 2012)". The age of bottom profiles in the studied mires with high peaks of Ti is about 8.5-7.0 ka (I take it from place Rybnaya mire, Fig. S5). So it coincided with period of drier climate conditions in Siberia. I think that flood events should be happen if precipitation increases, but precipitation was reduced at the period. How can you explain this discrepancy? I can imagine that this period of drier climate conditions might contribute to frequent fires, deforestation and enchanced soil erosion. This is just my opinion, but I think this is more reliable explanation for the Ti peaks, than river flood, that you suggested.*

R: The higher Ti content in the basal, minerogenic portion of Rybanya and, in fact, also at UC is not connected to floods but reflects the minerogenic substrate at the sites and therefore pre-dates the inception of the peat. We refer to subsequent (higher in the profile) Ti fluctuations as indicating possible flood events i.e., low frequency but high magnitude events which may not reflect the overall climatic trend at the time. As a lithogenic (or geogenic) indicator Ti can be seen as an indicator of detrital input reflecting the mineralogical content (in comparison to the highly organic nature most peat profiles). The source of this material will reflect both the type of mire and the events leading to this input. In an ombrotrophic context the input will be aeolian. Here the landscape position of the site means that (at this stage in the mire) a fluvial input (reflecting possible flooding or channel position change) is feasible as a transport mechanism for the delivery of such material and associated lithogenic signal (also seen in other lithogenic indicators but Ti has been selected as indicative). We have extended our discussion in the manuscript l. 347-352 and in the SI, File S5.

*Line 181 – you used the transfer function developed for the pan-European region (Amesbury et al. 2016) to derive the water depth from the studied peat cores. This transfer function was developed mainly (or even especially) for ombrotrophic and oligotrophic peatlands, but you applied this to the mesotrophic mires. I think this might increase the incorrectness of the reconstructed values of DWT in your study. Why did you not used the transfer function developed for Asian peatlands (Qin et al. 2021 in QSR), because the studied mires are located in Asia, but not in Europe? Apart from that, I can say that the transfer function developed for Asian peatlands includes more places with higher values of pH and therefore, I guess, it might be more suitable for reconstruction of DWT in the studied mires. Also I suggest adding testate amoeba diagrams from the studied peat cores to the Supplementary Materials of your manuscript. It is very important and interesting data. Furthermore, it would be very helpful to show the efficiency of the transfer function in your peat cores. There are standard statistic indexes (the chi square distance of fossil testate amoeba assemblage to the closest modern analogue from transfer function training set; goodness-of-fit statistic; the number of rare taxa and the number of absent taxa) indicating to what extent the fossil testate amoeba complexes in the cores correspond to the testate amoeba complexes embedded in the transfer function. We should avoid the situations when the half of taxa from fossil testate amoeba assemblages are absent in the taxon list of the transfer function and really do not contribute to the water depth reconstruction*

R: The study of Qui et al. (2021) was not available at time of running our quantitative DTW reconstructions and assemble of our datasets for this paper. However, the pan European transfer function (TF) is not limited to European sites but includes peatlands from Siberia and the sites having a large PH variation. Although, Qui et al. (2021) states that the Asian TF performs similarly to other large-scale TF, grouping of some taxa were less representative for our records then in the Amesbury et al (2016). Further, the paleohydrology reconstructions in the region using four transfer functions returned similar results (Willis et al., 2015). We will include a full comparison of the Siberian profiles using both training sets, representative diagrams for each site, and climate-based reconstruction in Diaconu et al. in prep.

*Lines 328-329 – you write "Warm summer temperatures likely enchanced evapotranspiration and consequently lowered peatland water levels, leading to drier surface conditions". You explain this for the period of "a temperature and moisture optimum between 6 and 4.5 ka BP (Groisman et al. 2012)" in Siberia. Although, if we look at the pollen diagrams from the studied mires (especially at Rybnaya mire Fig. S4a), we can see the increase of conifer pollen (Pinus sylvestris, P. sibirica) at this period and the decrease of Betula pollen. We can consider that conifers spread when precipitation exceeds evaporation (a prerequisite for the existence of the taiga). How can you explain this discrepancy when likely conifer trees indicate increased moisture (precipitation exceeds evaporation), but testate amoeba based DWT in mire indicate low levels (evaporation exceeds precipitation)? In general, I can conclude it looks very strange that*

*reconstructed water levels in your peat cores are not coincided directly to the climate changes, which you take from the monograph by Groisman et al. (2012) for Siberia. I think this is an additional argument that the reconstructed DWT values from the studied peat cores do not reflect hydroclimate changes of the study area.*

R: The temporal succession in the main tree taxa at our sites is in good agreement with other pollen records from western Siberia. There is also a good agreement between our and the few other charcoal records from the region in indicating a high fire activity between 8-4.5/ 5 ka. Wildfires occur predominantly during the growing season (sprint to autumn) but are most severe are during summer associated with dry soils or peatland conditions. Peatland conditions must have been dry at this time (8-4.5 ka), to allow such high fires to spread, and lower moisture conditions is also what TA reconstruction of depth to water level show. However, we have extended the discussion on lines 365-390.

*Along the manuscript you compare the forest composition and the DWT values reconstructed for mire. I think this does not make sense because DWT is related to mire condition, although the forest composition is related to forests (not to mire).*

R: Forest density and composition respond to changes in peatland hydrology. What is true, however, is that local hydrological conditions vary across peatland and this in turn is influencing the forest composition. In revising this manuscript, we introduced some words of cautious in linking the local variability in peatland hydrology with forest composition, please see, l. 432-450.

*Line 107 – when describing the "typical forests" for the study region you list tree composition of forested peatlands mainly and cite the researches "Berezin et al. 2014; Rybina et al. 2014", where peatlands (not forests) were studied. It is very strange to call these trees as light taiga, because in reality it is not forests, it is mire. I consider forested mire should not be called taiga. And why do you mention nothing about poplar, because it is one of the most abundant tree in forests from the study area?*

*R*: Thank you for this observation, please also see our response to Loiko, l. 112-116.

*Line 71 – you cite the work by Agee (1998), but I have not found this work in the Reference List. Please add this reference.*
*R*: Added

*Line 70 – Why does Betula pubescens belong to the group of invaders and to the group of endurers. In paper by Wirth (2005) the group of endurers includes B. pubescens from only northern taiga. Therefore it is incorrect to say that B. pubescens is endurer in your research.*
R: The fact that *B pubescens* can behave endurer only in northern taiga does not come out in Wirth 2005 However, we only mentioned its possibility to behave as endurer did not say this is the case of the study region.

*Line 555 - the title of the work is written twice in the link.*
R: Thank you, duplicate removed.

*Line 502 – in this reference 2020a is pointed after author's names. But in the next reference (Line 506) 2020b is pointed in the end of link or probably this is a part of DOI. What is the right variant of design? I think this detail should be uniform. If you have 2020a, then you have 2020b.*

R: The year of publication should be placed at the end of each citation. We have corrected this, thank you.

*Line 298 – Rudaya and coauthors in their paper (2020) studied two lakes from the Steppe Altai. Indeed, this is Altai, but it is not a mountainous region. Their study area is related to the southern part of West Siberian Plain (lowland). Please correct this phrase in your manuscript. By the way (Line 95) you cite this paper again as example of research conducted in Siberian boreal forests. It is incorrect, because steppe zone is not boreal.*
R: Thank you, corrected in both locations.

---

## Author Response (AR2)

Frankfurt am Main, 27.04.2022

Dear Editor, Nathalie Combourieu Nebout

Please find enclosed a revised version of the manuscript entitled "Holocene wildfire regimes in peatlands in western Siberia: interaction between peatland moisture conditions and the composition of plant functional composition".

We have addressed the major comments made by the reviewers, which include

1. running the Asian transfer function for testate amoeba and presenting the testate amoeba percentages into SI (File S5),
2. focussing on the relationship between forest composition and fire in section 4.3,
3. correcting any geographical misplacement of our comparisons with published records,
4. moderating the references to forested peatlands, including the removal of this term from the abstract

While we feel that these amendments were justified and have improved our manuscript, we consider now that we have reached the point where further revisions would be futile a fundamental difference in scientific opinion. Consequently, we ask that, as the Editor for our submission, you make a balanced judgement of this final resubmission.

Kind regards

Angelica Feurdean on behalf of all the co-authors

*Reviewer 2*
*There are 3 spelling errors on the citations of the publications of Barhoumi et al., 2019 and 2021: line 331, line 863, it is not "Bourhami", but "Barhoumi"*
R: Thank you we have corrected this.

*Reviewer Kurina.*
*I have read the revised version of the manuscript cp-2021-125 "Holocene wildfire regimes in forested peatlands in western Siberia: interaction between peatland moisture conditions and the composition of plant functional types" by A. Feurdean and coauthors. Unfortunately, I have not seen the essential corrections through the text made after the first cycle of review and comments that I expected to see. On the contrary, only some of the most disputable sentences that reviewers drew attention to were rewritten to smooth out sharp corners, after which these phrases became even more vague and uncertain, but the essence did not change despite the recommendations of the reviewers. For now as reviewer I decided to give the authors one more chance to correct the manuscript by making major, not minor, revisions. Now I list only global comments on the text of the revised manuscript. Although in addition I have found a series of minor errors and inaccuracies, which I will not mention now, because I think that after the recommended global rewriting of the manuscript, these inaccuracies may be removed occasionally and (or) new inaccuracies will appear in other places of the text. I am going to list minor errors only after major rewriting of the manuscript in according with my global comments. I repeat again, to my mind, this research contains valuable relevant precise data that are worthy to be published, although I disagree with interpretation of the results in this research. Therefore my global comments are aimed at improving the logical structure of the article and thoughtful interpretation of the data obtained in order to fit it objectively into the results from earlier publications about the study area.*

R: Although we agree with some of the points raised by the reviewer Irina Kurina, and have further revised the manuscript (see details below), we also feel that these comments reflect a lack of willingness to accept our interpretation and outline of the manuscript. This perspective contrasts with the other three reviewers who have all accepted the manuscript.

*1) I fundamentally insist on a change of the heading of the manuscript and data interpretation through the whole text. My main disagreement is that, in fact, authors do not investigate forested peatlands. They investigate vegetation cover (mainly tree composition) in boreal ecosystems taken together (without separating forested peatlands) in the selected study area. To reconstruct the history of vegetation cover at a long-term scale the authors take peat cores from two peatlands. This does not mean that they investigate peatlands, but this means*

*that they used a peatland as a natural archive for extraction the palaeoenvironmental information. By the way, the authors said this idea by themselves in the Introduction section (see Line 98). Furthermore, to reconstruct the history of vegetation cover the authors apply pollen analysis, which reflects regional vegetation in general, including both forests and peatlands. The only possibility for the reconstruction of vegetation directly in the peatland, I consider through the analysis of plant macrofossils in peat as it was performed in the research by Magnan et al. (2012) (this paper is referred in the manuscript). The authors did not analyze the plant macrofossils, so they cannot reconstruct the vegetation in the peatland, but only the vegetation of the region as a whole, and they cannot separate objectively between pollen from forest and pollen from forested peatland. I emphasize the importance of this comment on the example of the research by Mikhailova et al. (2021) (this paper is referred in the manuscript), in which, for the same peat sequence, obvious differences between the results of the plant macrofossil analysis and the pollen analysis were demonstrated. In the revised manuscript (lines 439-440) the authors postulate that most of the tree pollen deposited on the surface of the studied peatlands comes from trees in the peatlands, because forested peatlands are widespread in West Siberia. I principally disagree with this statement. In the previous comments Irina Kurina asked the authors to display the definite figures of the area, occupying by peatland ecosystems, which are available from the recommended publications. As a result the authors added the references to these publications, but they have not displayed the certain figures of the area with peatlands. Therefore I point these figures here, based on the following papers (Vompersky et al. 2011 in Contemporary Problems of Ecology; Kremenetski et al. 2003 in QSR; Liss et al. 2001 monograph Wetland systems of Western Siberia and their conservation value – in Russian; Alekseeva et al. 2015 in Bulletin of Tomsk Polytechnic University – in Russian). In the southern taiga of West Siberia forests occupy 50% of the area, peatlands occupy 30%, including forested peatlands (15%) and open peatlands (15%). It means that the area of forests (50%) more than three times greater than the area of forested peatlands. It follows from this that tree pollen from forests prevails in general pollen rain, rather than tree pollen from forested peatlands. Thus, again I strongly recommend to add into the manuscript the certain figures about area occupied by peatlands, both forested and open, and by forests (for comparison) in the study region of the southern taiga of West Siberia. And I urge the authors to correct their assumption (lines 439-440) that most arboreal pollen in their spectra is derived with trees from forested peatlands. The figures show that this assumption is wrong or at least is doubtful.*

R: We have removed the words 'forested peatland' from the title and in most places throughout the revised text. While we are aware that our pollen data records the mixed composition of trees on the peatland and surrounding forest, we disagree that our reconstruction merely used peatland archives to reconstruct vegetation. Firstly, pollen gives a regional picture of vegetation, but the spatial scale of the reconstruction is strongly influenced by tree cover and canopy density. Given the forested landscapes surrounding the study sites (see Fig.1), it is likely that that pollen from the immediate neighbourhood of the coring sites dominate the sequences. Secondly, the percentages of forests, forested and non-forested peatlands mentioned by the reviewer refer to western Siberia, whereas Fig. 1 shows that at the study sites. The combination of the two points supports that a significant part of the reconstructed tree cover comes from forests on the peatlands. We have added the numbers requested and revised lines 119-122.

*2) The authors pointed water table depth of 20 cm as threshold determining correlation between peatland surface moisture and severity of the fires (see lines 46 and 347). I want to note that this value in itself (water table depth equal to 20 cm) does not carry ecological meaning. Because this figure is a result of calculations made by transfer function, which is based on a training set used one-off measurements of water table depth made during field work in different mires in different periods of the growing season (different months). Moreover this value of reconstructed water table depth is obtained for the peat sample, including a mixture of testate amoebae for at least 1-2 dozens of years (it depends on time resolution for the peat core), not for a one year. Therefore logically it is not correct to compare this value with modern one-off or mean annual values of water table in peatlands. And please note that mean annual measurements now are made for sparse single peatlands. Thus, I want to emphasize that the results of testate amoeba-based water table reconstruction are used to show the dynamics of water table changes. For example, based on these values we can say that surface wetness in the peatland becomes dryer or wetter, but we should not focus on definite values like water table changed from 20 to 10 cm. I foresee that in other peatlands the other individual thresholds will be obtained, for example, 10 cm or 25 cm or any other, while in this research the threshold of 20 cm was calculated. Therefore I recommend removing the value 20 cm from the Abstract (line 46). Instead, for Abstract, the authors might say that they revealed the statistically significant or quantitative relationship between fire severity and peatland moisture.*

R: We have removed the threshold values in water table depth and the statistical correlation of absolute water table depth and charcoal (GLM model) from the entire manuscript and figures (Fig. 6 c,d). In the light of new results for the DTW values, we have replaced the absolute values (cm) with standardized values (see lines 200-205; 282-290).

*3) Because the authors have no data on the composition of trees exactly in the forested peatlands, it makes no sense to compare directly tree composition and peatland moisture. Therefore, please, remove such comparison through the text of the manuscript. I consider the arboreal pollen data in the research mainly reflects tree composition in surrounding forests, although water table in the forested peatlands differs from water table in forest ecosystems.4) The authors are aimed to study the interactions between climate, fires, tree composition of forested peatlands and peatland moisture. I will comment this logical sequence in detail: Climate. In fact, the authors did not reconstruct regional climate changes and did not make an attempt to fit their research to the available information about regional climate from the earlier publications (as example, Borisova et al. 2011; Groisman et al. 2013 – referred in the manuscript). And this looks strange, because the authors received the pollen data and might apply this to climate reconstruction, at least, as qualitative description, not quantitative estimation of climate parameters. Moreover, please, note that available regional palaeoclimate information are based on pollen data (Borisova et al. 2011; Groisman et al. 2013). If the authors do not reconstruct regional palaeoclimate using the pollen data, they should at least to compare their pollen diagrams obtained with the other earlier published pollen diagrams from the study region. It is especially important, because possible similarity with other pollen records will allow concluding that pollen data surely reflect a common regional signal of vegetation succession and regional palaeoclimate changes. I strongly recommend to do this and to discuss the results of this comparison in the Discussion section of the manuscript. Furthermore, to my mind, your pollen records from the two studied peatlands are too different to be combined into one composite sequence. This also applies to data on the reconstructed water table depth and even to the fire dynamics. I think, it is better to consider the data from the two studied peatlands separately and to highlight their dissimilarity between each other, rather than similarity. I suppose that two studied sites (Rybnaya and Ulukh-Chayakh) related to different climatic regions (subregions), despite they are 200 km apart. In fact, if the authors compare their pollen records with the available pollen records from the studied area, they will notice that pollen record from the Rybnaya peatland is more similar with pollen records located north-westward in the southern taiga of West Siberia (please see the pollen diagram from Bugristoe mire in Blyakharchuk and Sulerzhitsky 1999; pollen diagram called Tom-river-mouth in review by Zang and Feng 2018). In these diagrams, note the clear positive peak of Abies pollen (or at least other dark coniferous pollen) in the Early Holocene (between 7.0-4.5 ka BP). The other pollen record from the Ulukh-Chayakh peatland is more similar to pollen diagrams taken southwards and south-eastwards (as examples, diagram from the Kirek lake and Chaginskoe peat in Zhang and Feng 2018; Teguldetskoye peatland in Blyakharchuk 2012; Zhukovskoe peat in Borisova et al. 2011; Pinchinskoye mire in Mikhailova et al. 2021). In these pollen records, there is clear positive peak of Abies pollen in the Mid Holocene (between 4.0-2.5 ka BP), although the peak of Abies pollen is absent between 6.0-5.0 ka BP. I recommend the authors to read the papers by T. Blyakharchuk and other palynologists and to pay attention to the principles of climate reconstruction based on pollen data. You can see that for the southern boundary of taiga belt in West Siberia an expansion of boreal forests to the south is limited mainly by humidity of the climate, rather than by temperature, unlike for the northern boundary of taiga belt. It follows from this that increase of precipitation amount causes movement of southern taiga to the south, while decrease of precipitation leads to the retreat of the taiga to the north. In the pollen record taken near the southern boundary of taiga belt this pattern appears as increase of dark coniferous tree pollen (considered as greater climate humidity) alternating with increase of Betula pollen (considered as less climate humidity). This means that increase of dark coniferous tree pollen indicates greater climate humidity. To my mind, peaks of dark coniferous tree pollen are not synchronous in your two records from Rybnaya and from Ulukh-Chayakh. This difference highlights the difference in local climatic conditions at these two places. Also pay attention to the dynamics of flooding events in the Holocene demonstrated in the research by Mikhailova et al. 2021 (this work is referred in the manuscript, see Fig. 9 there in). You can see that great floodings (considered as reflection of increased climate humidity) were observed at different periods in the Holocene for the records from Siberian taiga and from more southern forest-steppe. Fires. I would underline a similarity between fire severity and trees composition revealed by pollen data. At the Rybnaya record most severe fires were observed at 7.5-6.0 ka and at 4.5 ka BP, in the both these periods severe fires correspond with increasing of dark coniferous tree pollen (Picea, Abies, Pinus sibirica); while at the Ulukh-Chayakh place severe fires were documented before 6.0 ka and at 3.5 ka BP, in the first period this may corresponds with increased abundance of Picea pollen (not yours, but regional pollen data from earlier publications), in the second period fire peak corresponds with increase of Picea and Abies pollen in the Ulukh-Chayakh record. Pay more attention to the research by Barhoumi et al. (2021) (this is referred in the manuscript). The authors found a correspondence of fire dynamics in this research at the Baikal Lake region to the own research. Barhoumi and coauthors claimed they expected to find more severe fires at drier climate conditions, rather than at wetter climate. Although as a result they concluded that the severity of fires in the Holocene is apparently related to the tree composition of forests (more exactly the crown structure of different trees), rather than to climate conditions. This conclusion was mainly based on the fact, that in the Early Holocene they reconstructed the most moisture palaeoclimate, coincided with increase of dark coniferous tree pollen and with*

*period of severe crown fires. So I recommend the authors of the manuscript to make similar conclusions that clearly follow from the results of the research: the fire severity is more influenced by the tree composition of the forest than by climatic conditions. Moreover, the authors can enhance these conclusions if they follow the next logic sequence: increase of climate humidity leads to the spread of dark coniferous trees, which facilitate, in turn, more severe (crown, not surface) fires. Tree composition. I have explained already that the authors cannot consider separately tree composition in forested peatlands, although they can discuss noteworthy interactions between tree composition of boreal ecosystems taken together and fire severity.*

R: 4.3 focuses now on the relationship between PFTs and fire regime (most of the feature highlighted by the reviewer have already been addressed), with only a slight touch on the possible relationship between the water table and dominant PFTs. It is beyond the aims of this study to provide a pollen-based climate reconstruction or large -scale synthesis of pollen diagrams. Most (none) of the suggested pollen diagrams have no charcoal counted; thus, an extensive comparison between various pollen records will not strengthen the fire-vegetation relationship but merely provide similarities/divergences between vegetation/forest composition at a large scale, which is outside the scope of our study. However, we have included the recommended references. (Please see 4.3). It should be noted that composite pollen, charcoal, and water-table have been widely used to obtain regional pictures of vegetation, fire, and water table level beyond local trends.

*Peatland moisture. In addition to the recommendations above, I propose to strengthen the main conclusions with data on the moisture on the peatland surface. I have mentioned already that peatlands (both forested and open) occupy 30% of the area in the southern taiga of West Siberia. It might be concluded that in one case when the peatland moisture increases (in this research the reconstructed water table depth less than 20 cm), peatlands do not burn themselves and prevent the spread of fires in surrounding forests. Thus peatlands localize and stop fires, which in most cases are initiated in forests (not in peatlands). In other case when the peatland moisture decreases (in this research the reconstructed water table depth exceeds 20 cm), fires may attack peatlands and burn them. Thus, in the contrary, peatlands become fire spreaders*

R: 4.2.1 The influence of peatland moisture on fuel type and flammability We have removed the threshold values in water table depth. We have also strengthened the link between peatland moisture and probability of fire (l. 347-353).

*In the revised manuscript the authors made an attempt to compare the results of water table depth reconstructions, based on testate amoeba data. I think this comparison is extremely incorrect. Lines 377-378 – the authors mentioned drier period between 7.5 and 5.5 ka BP and referred to Kurina et al. (2018) and Kurina et al. (2021). I insist on clarification that in the research by Kurina et al. (2018) aggregated (combined from different records) drier period was observed between 7.0 and 6.0 ka BP. I have not found the reference Kurina et al. (2021), although there is reference of Mikhailova et al. (2021), where the drier period was revealed between 7.5 and 5.1 ka BP. BUT this research is not related to the taiga zone of West Siberia. This is forest-steppe zone of East Siberia. Line 374 – the authors wrote that the research by Mikhailova et al. (2021) was conducted in West Siberia. Again, this is wrong. This work is related to East Siberia. Lines 385-387 – the authors registered a similarity of their research with other investigations from West Siberia. Here they referred to Mikhailova et al. 2021 and to Blyakharchuk and Kurina (2021). Again, this is wrong assumption. I have said already about research by Mikhailova et al. (2021). Moreover, the research by Blyakharchuk and Kurina (2021) is carried out in southern Central Siberia (not West Siberia) and in the mountain region (not the part of West Siberian Plain). Next, for these two records the authors mentioned similar wet period at 5.1-1.4 ka BP. This period is represented completely only in the work by Mikhailova et al. (2021), while the record by Blyakharchuk and Kurina (2021) covers only the last 2500 years, so it cannot represent adequately a peatland moisture between 5.1 and 1.4 ka BP. I consider, all the mentioned regions (southern taiga of West Siberia, forest steppe of East Siberia and mountains of southern Central Siberia) have specific regional climate conditions. I suppose that all the coincidences between these records are probably occasional. In addition, to my mind, these records are as similar as they are dissimilar. In other words the similarity between these records is subjective and doubtful. In general, I tend to expect, with an increase in climate humidity, the surface wetness of peatlands also increases, to which testate amoebae response; and vice versa. In quantitative reconstructions, it is customary to calculate the water table depth according to testate amoeba data, although in reality testate amoebae respond to surface wetness, rather than to water table depth in itself. Based on this, it is not clear to me, why in the Rybnaya record for the period at 7.5-4.5 ka BP drier conditions on peatland surface were reconstructed, while the pollen record demonstrated increase of dark coniferous tree pollen indicating more humid climate. In the Ulukh-Chayakh place the picture is even more complicated. As example, in the period at 5.0-2.5 ka BP pollen data displayed an increase of dark coniferous tree pollen (considered as more humid climate conditions), while the reconstructed water table depth reflected drier conditions on peatland surface between 5.0 and 3.5 ka BP and then wetter conditions between 3.5 and 2.5 ka BP. To my mind, this looks very strange and unclear. Unfortunately, the authors inflexibly refuse*

*to show the initial data on the distribution of testate amoeba taxa in their peat sequences, on the basis of which the reconstruction of the water table depth was made. So, as reviewer, I have to blindly appreciate how correctly the authors calculated water table depth in their peat sequences. In this case, I have no reason to confirm the correctness of the calculations. Moreover unexpected discrepancy between pollen and testate amoeba data makes me suspect possible errors in the calculations of water table depth. Now I cannot judge surely. Therefore I have to insist that the authors show additional evidence of the correctness of the calculations of the water table depth in the peat sequences.*

R: There are two independent transfer functions (TF) showing similar trends in hydrological conditions at all four sites. The Asian TF includes calibration sites from Siberia. In File SI 5, you can view the results of the two TF and the complete list of testate amoeba percentages. We do apologize for misplacing some of the TA records; this was corrected in the revised paper. The timing and directions of the hydrological shifts were accurately presented. Please note that Mikhailova et al. (2021) paper also presents a synthesis of palaeohydrological records from the southern part of western Siberia to which we made our comparisons. Generally, there seems to be divergent trends in moisture conditions reflected by pollen versus other hydrological proxy for the 8-4.5 ka.

*For reconstruction of water table, the authors use the pan-European transfer function (Amesbury et al. 2016), which is based on training set from the European area. Not, it is important; this training set does not include samples from Siberia territory. Among experts on testate amoebae, the question remains whether it is correct to apply the transfer function developed for one region to a peat sequence from another region. Therefore the testate amoeba specialists recommend using the transfer function from the same region. Fortunately, for the area you are studying, several different transfer functions have been developed: pan-Asian transfer function by Qin et al. (2021) and three local transfer functions by Kurina and Li (2019) developed from ombrotrophic, minerotrophic and combined mixture of mires. I urge the authors to reconstruct the water table depth in their peat sequences using the different transfer functions proposed and to demonstrate the results from different transfer functions are similar with the results from the pan-European transfer function. I consider it is incorrect to state that if Willis et al. in their work (2015) applied four transfer functions from other regions to the Siberian peat sequences (because at that time they did not find a transfer function covering the Siberian territory!) and obtained similar reconstruction results, then the authors will get same way. An additional complication is that, in contrast to the Willis et al.'study (2015), which applied transfer functions developed for raised bogs to peat sequences from raised bogs, the authors in this study use a pan-European transfer function developed for ombrotrophic peatlands for peat sequences from mesotrophic peatlands. Meanwhile the research by Kurina et al. (2020) evidenced that the application of the transfer function for ombrotrophic peatlands to peat sequences including minerotrophic peat can lead to significant distortions of the reconstructed water table depth compared to the results of reconstruction using the transfer function for minerotrophic peatlands. Do not say that other transfer functions are not available for you. The truth is that they are available. You can email the authors of the pan-Asian transfer function and they help you to make reconstruction. All necessary data on the three transfer functions by Kurina and Li (2019) are free and uploaded at Mendeley DataBase. You can find it using the combination of the author name (Kurina) and key words (testate amoebae) and apply to your peat sequences by yourselves. I know that the calculation of water table depth is easy and fast (it will take no more than one working day from you). In addition, I ask the authors to represent in the manuscript or in the supplementary materials the certain figures of summarized relative abundance by those testate amoeba taxa from each peat sample, that were really involved into the calculation of water table depth, i.e. these taxa are present in the training set of the transfer function. These figures will help to reveal the transfer function, which mostly corresponds with your peat sequences. And representation of these figures will be more objective argument to confirm the suitability of transfer function to peat record, rather than only saying this is suitable and this is not suitable without any evidence as the authors did after the first cycle of review-*

*Corects wets Siberia !L374 this sentence does not placed Michaolve study in wetsner Siberia but say tata Lines 377-378 – the authors mentioned drier period between 7.5 and 5.5 ka BP and referred to Kurina et al. (2018) and Kurina et al. (2021). I insist on clarification that in the research by Kurina et al. (2018) aggregated (combined from different records) drier period was observed between 7.0 and 6.0 ka BP. I have not found the reference Kurina et al. (2021), although there is reference of Mikhailova et al. (2021), where the drier period was revealed between 7.5 and 5.1 ka BP. BUT this research is not related to the taiga zone of West Siberia. This is forest-steppe zone of East SiberiaLine 374 – the authors wrote that the research by Mikhailova et al. (2021) was conducted in West Siberia. Again, this is wrong. This work is related to East SiberiaLines 385-387 – the authors registered a similarity of their research with other investigations from West Siberia.*

R: We have run two independent transfer functions, one developed for the pan-European region (Amesbury et al., 2016) and the other for Asia (Qin et al., 2021). The Asian TF includes calibration sites from Siberia. Following literature recommendations (Amesbury et al., 2016; Swindles et al., 2016; 2019; Qin et al., 2021), we standardised

the reconstructed water-table depth values to Z scores.  The two independent transfer functions (TF) show similar trends in hydrological conditions at all four sites; therefore, this revision does not impact our previously major interpretation of the hydrological conditions In File SI 5, one can view the results of the two TFs and the complete list of testate amoeba percentages (see more see l. 200-204, l. 282-290) and apply as many TFs as pleased.